# Bioactive potentiality of secondary metabolites from endophytic bacteria against SARS-COV-2: An *in-silico* approach

Yasmin Akter[1‡], Rocktim Barua[1‡], Md. Nasir Uddin[1], Abul Fazal Muhammad Sanaullah[2], Lolo Wal Marzan[1]*

**1** Department of Genetic Engineering and Biotechnology, Faculty of Biological Sciences, University of Chittagong, Chittagong, Bangladesh, **2** Department of Chemistry, Faculty of Sciences, University of Chittagong, Chittagong, Bangladesh

‡ These authors contributed equally to this work and share the first authorship.
* marzan.geb@cu.ac.bd

**Data Availability Statement:** We declare that our manuscript data are fully available without restriction.

## Abstract

Five endophytic bacterial isolates were studied to identify morphologically and biochemically, according to established protocols and further confirmed by 16S rDNA Sanger sequencing, as *Priestia megaterium*, *Staphylococcus caprae*, *Neobacillus drentensis*, *Micrococcus yunnanensis*, and *Sphingomonas paucimobiliz*, which were then tested for phytohormone, ammonia, and hydrolytic enzyme production. Antioxidant compounds total phenolic content (TPC), and total flavonoid content (TFC) were assessed by using bacterial crude extracts obtained from 24-hour shake-flask culture. Phylogenetic tree analysis of those identified isolates shared sequence similarities with the members of *Bacillus*, *Micrococcus*, *Staphylococcus*, and *Pseudomonas* species, and after GenBank submission, accession numbers for the nucleotide sequences were found to be MW494406, MW494408, MW494401, MW494402, and MZ021340, respectively. *In silico* analysis was performed to identify their bioactive genes and compounds in the context of bioactive secondary metabolite production with medicinal value, where nine significant bioactive compounds according to six different types of bioactive secondary metabolites were identified, and their structures, gene associations, and protein-protein networks were analyzed by different computational tools and servers, which were reported earlier with their antimicrobial, anti-infective, antioxidant, and anti-cancer capabilities. These compounds were then docked to the 3-chymotrypsin-like protease (3CL^pro) of the novel SARS-COV-2. Docking scores were then compared with 3CL^pro reference inhibitor (lopinavir), and docked compounds were further subjected to ADMET and drug-likeness analyses. Ligand-protein interactions showed that two compounds (microansamycin and aureusimine) interacted favorably with coronavirus 3CL^pro. Besides, *in silico* analysis, we also performed NMR for metabolite detection whereas three metabolites (microansamycin, aureusimine, and stenothricin) were confirmed from the 1H NMR profiles. As a consequence, the metabolites found from NMR data aligned with our *in-silico* analysis that carries a significant outcome of this research. Finally, Endophytic bacteria collected from medicinal plants can provide new leading bioactive compounds against target proteins of SARS-COV-2, which could be an effective approach to accelerate drug innovation and development.

**Funding:** The author(s) received no specific funding for this work.

**Competing interests:** The authors have declared that no competing interests exist.

## Introduction

Endophytes are symbiotic microorganisms that thrive in seemingly healthy interior plant tissues without causing any harm to the host plants, and they are mostly bacterial and fungal species [1]. Mutualism has become synonymous with endophytes, as both hosts and microorganisms exhibit a synergistic relationship with each other. To date, the symbiotic interaction between plants and endophytes appears to be beneficial to both sides: it benefits the endophyte by increasing the availability of the plant's nutrients [2], and it benefits the plant by providing pathogen protection, improving nutrient uptake, promoting plant growth, and stress tolerance [1–3]. Natural compounds from endophytes exhibit activity as antimicrobials, antifungals, anticarcinogens, immunosuppressants, or antioxidants [4]; thus, secondary metabolites produced by endophytes can be implemented as therapeutics in the pharmaceutical and agricultural industries [3]. Endophytes have been linked to more than 200 genera from 16 phyla of bacteria [5], with the majority of the species belonging to the phyla Proteobacteria (90%) and most regularly Firmicutes and Actinobacteria [6, 7]. The diversity of endophytic bacteria ranges from gram-positive to gram-negative bacteria throughout all ecosystems [8]. Bacterial endophytes occupy an ecological niche comparable to phytopathogens, making them good candidates for biocontrol agents. Plant growth-promoting (PGP) endophytes have favorable influences on plant growth because of their metabolic activity and functional variety [7, 9]. They promote plant growth by developing symbiotic relationships, nitrogen fixation, phosphate solubilization, and the generation of important phytohormones such as indole acetic acid (IAA), abscisic acid, cytokinin, and others [10, 11]. Secondary metabolites are organic chemicals produced by plants, bacteria, or fungi that are not directly involved in the organism's regular growth, development, or reproduction, known as specialized metabolites, toxins, secondary products, or natural products [12]. It is not required for bacteria to flourish, but it does help them to interact more effectively with their environment. Bacterial endophytes produce terpenoids, alkaloids, polyketides, nonribosomal peptides (NRPs), phenols, enzymes, and phytohormones that aid plant-bacteria interactions and colonization.

Endophytic bacteria from medicinal plants offer the possibility of discovering a wide range of chemicals as well as a sustainable source of natural products. It has been reported that in medicinal plants, Bacillales, Enterobacterales, and Pseudomonadales were the most prevalent orders, accounting for 72.62% of the total, and *Bacillus*, *Pantoea*, and *Pseudomonas* were the most prevalent genera, accounting for 58.92%. *Bacillus*, *Pseudomonas*, and *Paenibacillus* can influence the growth, stress resistance, and metabolism of medicinal plants, and Streptomyces has been observed to promote plant growth and development [13]. Surfactin, Iturin, Fengycin, Munumbicins, and many more bioactive compounds with anti-infective, antimicrobial, anticancer, and anti-inflammatory bioactive compounds have been purified from various medicinal plants throughout the world [14].

Genome sequencing, comparative genomics, microarray, next-generation sequencing, metagenomics, and metatranscriptomics are some of the techniques utilized or that can be employed in modern endophytic research to unravel the plant–endophyte connection. Through the genome mining approach, the genetic characteristics that directly or indirectly govern diverse bioactivities, as well as putative bioactive secondary metabolites, have been discovered. It helps to identify specific genes involved in antibiotic resistance mechanisms, antibiotic synthesis, plant growth promotion, the endophytic secretory system, surface attachment and insertion elements, the transport system, and other metabolic systems [15]. On the other hand, comparative multigenome analysis is very useful in understanding the genetic and metabolic diversity of comparable or related microorganisms that interact with plants and animals in various ways [16]. To identify the gene clusters (smBGCs) for the secondary metabolite

biosynthetic pathways in the genome, several bioinformatics tools have been established, including BAGEL [17], ClustScan, CLUSEAN [18], NP searcher, PRISM [15], and antiSMASH [19]. The smBGC database has been set up to aid the search for known and novel metabolites that can be linked to traditional ways of characterizing molecules through chromatography and spectroscopy. Currently, *in silico* analysis has created a junction to predict and validate laboratory and computerized analyses [20].

Microbial secondary metabolites (MSMs) have been utilized to synthesize novel medicines to treat a variety of pathogens, including viruses, bacteria, fungi, and parasites, since they are simple and dependable sources [21, 22]. These secondary metabolites can be detected by different techniques such as NMR, GC, LC, CE etc [23]. Nuclear magnetic resonance (NMR) spectroscopy has been recognized as a reliable method that frequently applied in metabolomics studies [24]. A variety of metabolites can be identified by using 1H NMR spectroscopy as well as different bioactive antiviral compounds [25].

The antiviral bioactive compounds isolated from different bacteria are listed in **Table 1**. The recent pandemic officially known as coronavirus disease 2019 (COVID-19) is caused by a novel coronavirus known as severe acute respiratory syndrome coronavirus 2 (SARS-COV-2) that poses a threat to global public health.

It was initially detected in December 2019 in Wuhan, China [26], and on 11 March 2020, the outbreak was declared by the World Health Organization (WHO) as a global pandemic that has caused a total death of 4,644,740 worldwide as of September 15, 2021, with nearly 4 million new cases reported globally in the past week (6–12 September) according to the World Health Organization. Most *in silico* research is ongoing around the world to identify anti-SARS-COV-2 drugs of either microbial or plant origins. Diversified microbial metabolites have been depicted to represent promising antiviral activity against a variety of DNA and RNA viruses, which is turning to an emerging trend for therapeutic applications of microbial metabolites as antiviral agents [27]. Therefore, metabolites produced by endophytic microorganisms might act as an emerging source of antiviral bioactive compound [1, 28–30]. Molecular docking and other computing approaches have been useful in the first large-scale screening of numerous natural chemicals and small molecules that directly inhibit critical target proteins in this direction. Reports on virtual screening of current

**Table 1. Antiviral bioactive compounds isolated from bacteria.**

| Microorganisms | Antiviral compounds | Group | Virus/Disease |
|---|---|---|---|
| *Streptomyces avermitilis* | Avermectin B1a | Lactone | NR |
| *E. coli* | Baicalein | Lactone | DENV-2, SARS-COV2 |
| *E. coli* | Scutellarein | Lactone | SARS-COV |
| *E. coli* | Rosmarinic acid | Lactone | HCV, HIV |
| *Lactobacillus sp* | Exopolysaccharide | Polysaccharide | Adenovirus |
| *Streptomyces koyangensis* | Neoabyssomicin D | Polyketone | HSV |
| *Streptomyces sp* | Antimycin A | NR | Western equine encephalitis virus |
| *Streptomyces roseus* | Leupeptin | Peptide | Marburg virus |
| *Bacillus licheniformis* | Exopolysaccharide | Polysaccharide | HSV1 |
| *Myxococcus stipitatus* | Phenalamide | Peptide | HIV-1 |
| *Bacillus pumilus* | Pumilacidins A-G | Peptide | HSV1 |
| *Labilithrix luteola* | Labindoles A; Labindoles B | NR | HCV |

**Note: HIV**: Human immunodeficiency virus; **HSV**: Herpes simplex virus, **HCV**: Hepatitis C virus, **DENV**: Dengue virus, **SARS-COV**: Severe acute respiratory syndrome coronavirus

antiviral medicines based on known knowledge of SARS-COV-2 and closely similar corona-viruses, available databases and natural agents against emerging targets such as viral spike proteins, envelope protein, protease, nucleocapsid protein, and 3CL hydrolase are rapidly emerging [31–33]. The 3-chymotrypsin-like protease (3CL$^{pro}$), which is well known as the main protease (M$^{pro}$), and the papain-like protease (PL$^{pro}$) of the virus are two proteases vital to transcription and replication of the proteins encoded by the viral genome due to their peculiarity in splitting the two translated polyproteins (PP1a and PP1ab) into separate functional constituents [34]. A variety of flavonoids have been reported to have antiviral activity against SARS-COV by inhibiting 3C-like protease (3CL$^{pro}$) [35], where the 3CL$^{pro}$ of COVID-19, which is known as the main protease, depicted 96% sequence similarity with that of SARS-COV [36]. As the autocleavage process is very important for viral propagation, 3CL$^{pro}$ acts as an excellent drug target for anti-coronaviral infection. Therefore, 3CL$^{pro}$ can be considered an excellent drug target, and many efforts have been assigned to its study, as it has a key role in the replication cycle [33].

Therefore, the aims of the present study are to isolate and characterize endophytic bacteria from different parts of medicinal plants, evaluate their plant growth-promoting and antioxi-dant properties, predict and detect their secondary metabolites and analyze bioactive metabo-lites as potential antiviral compounds against SARS-COV-2.

## Materials and methods

### Sample collection

A total of 16 fresh and healthy medicinal plants were collected from the University of Chitta-gong campus (Table 2) into sterile zipper bags and then labeled and sealed. The samples were then immediately transferred to the lab maintaining a cold chain and stored in a 4˚C refrigera-tor for further processing.

**Table 2. List of collected medicinal plant samples and their collection areas.**

| Serial No. | Date of Collection | Season | Name of Plants | Location (Around CU campus) | Types of Plants | Plant Parts Used | Isolate code |
|---|---|---|---|---|---|---|---|
| 1 | 15/07/2019 | Rainy | *Ocimum tenuiflorum* (তুলসি) | 1 no. Gate | Herb | L, R, S | TL, TR, TS |
| 2 | 18/07/2019 | Rainy | *Eclipta prostrata* (কালসো·না) | Science Faculty | Herb | L, R, S | KL, KR, KS |
| 3 | 28/07/2019 | Rainy | *Justicia adhatoda* (বাসক) | Central Mosque | Shrub | L, R, S | BL, BR, BS |
| 4 | 03/08/2019 | Rainy | *Clerodendrum viscosum* (ভাট) | Biol. Sci. Faculty | Shrub | L, R, S | VL, VR, VS |
| 5 | 03/08/2019 | Rainy | *Leonurus sibiricus* (দ্রো·ণপুষ্পী) | Biol. Sci. Faculty | Herb | L, R, S | DL, DR, DS |
| 6 | 03/08/2019 | Rainy | *Melastoma malabathricum* (দাঁতরাঙা) | Biol. Sci. Faculty | Shrub | L, R, S | DaL, DaR, DaS |
| 7 | 14/10/2019 | Autumn | *Centella asiatica* (থানকুনি) | Near Rail Gate | Herb | L, R, S | ThL, ThR, ThS |
| 8 | 14/10/2019 | Autumn | *Catharanthus roseus* (নয়নতারা) | Central Mosque | Herb | L, R, S | NL, NR, NS |
| 9 | 14/10/2019 | Autumn | *Terminalia arjuna* (অর্জুন) | Alaol Hall | Tree | L, R, S | ArL, ArR, ArS |
| 10 | 05/02/2020 | Winter | *Mimosa pudica* (লজ্জাবতী) | Biol. Sci. Faculty | Herb | L, R, S | LL, LR, LF |
| 11 | 05/02/2020 | Winter | *Mikania micrantha* (জার্মান লিতা) | Biol. Sci. Faculty | Herb | L, R, S | GL, GR, GS |
| 12 | 07/04/2021 | Summer | *Terminalia chebula* (হরতকি) | Biol. Sci. Faculty | Tree | L, R, S | HL, HR, HS |
| 13 | 10/04/2021 | Summer | *Phyllanthus emblica* (আমলকী) | Biol. Sci. Faculty | Tree | L, R, S | AL, AR, AS |
| 14 | 18/04/2021 | Summer | *Mentha spicata* (পুদিনা) | Zero Point | Shrub | L, R, S | PL, PR, PS |
| 15 | 23/04/2021 | Summer | *Terminalia bellirica* (বহেড়া) | 2 No. Gate | Tree | L, R, S | BoL, BoR, BoS |
| 16 | 23/04/2021 | Summer | *Azadirachta indica* (নিম) | 2 No. Gate | Tree | L, R, S | NeL, NeR, NeS |

Note: Leaf: L; Root: R; Stem: S. Biological Sciences Faculty: Biol. Sci. Faculty.

## Isolation of endophytic bacteria

The total procedure was performed according to Ferreira et al., 2008, where 48 collected plant parts (2–3 cm length) were cleaned under running tap water to remove debris and then air dried [37]. Surface sterilization was carried out by rinsing them with Tween-20 for 10 minutes, followed by further washing with $dH_2O$ at least 7 times. After that, samples were dipped into 70% alcohol for 30 seconds and then washed with $dH_2O$. Twenty (20.0) ml of 0.2% $Hg_2Cl_2$ solution was added to the samples in a beaker, which was placed on a shaker at 240 rpm for 5 minutes at 27˚C. Then, the samples were washed again with $dH_2O$ at least 7 times. The final samples rinsed water was used as a control and spread onto nutrient agar plates [38], which contained (g/L)—peptone 5.00, beef extract 2.00, yeast extract 3.00, NaCl 5.00 and agar 18.00, where the pH was adjusted to 7.0. For the isolation of endophytic bacteria, samples were further treated in sterile phosphate-buffered saline (PBS) [39] containing (g/L) NaCl 8.00, KCl 0.20, $Na_2HPO_4$ 1.44 and $KH_2PO_4$ 0.24, where the pH was adjusted to 7.4 and maintained at 28˚C under 50 rpm agitation. All plates, including the control, were incubated at 37˚C for 5 days, and the number of CFUs was determined (**Table 3**) to estimate bacterial population density according to Addisu and Kiros, 2016 [38]. Following purification, morphologically distinct colonies were identified by observing colony characteristics such as gram nature, color, and shape using a binocular biological microscope (XSZ-107BN), where colonies of similar morphological features were grouped into the same species [40, 41]. Thus, isolates were selected, cultured, purified and stored in the laboratory at -80˚C in glycerol stock (50%) solution for further studies.

## Phenotypic and biochemical characterization of endophytic bacterial isolates

Standard tests for morphological and biochemical analysis were performed for the identification of endophytic bacteria. They were characterized by Gram staining and biochemical tests as described in the Cowan and Steel's Manual for the identification of medical bacteria [42]. For the activities of oxidase, catalase, citrate utilization, indole production, methyl-red (MR),

**Table 3. List of isolated bacterial isolates and host medicinal plants.**

| Serial No. | Sample Code | Host Medicinal Plant | Plant parts used |
|---|---|---|---|
| 1 | GL | *Mikania micrantha* | Leaf |
| 2 | GR | | Root |
| 3 | LL1 | *Mimosa pudica* | Leaf |
| 4 | LL2 | | Leaf |
| 5 | LF | | Stem |
| 6 | HL1 | *Terminalia chebula* | Leaf |
| 7 | HL2 | | Leaf |
| 8 | HS1 | | Stem |
| 9 | HS2 | | Stem |
| 10 | AL1 | *Phyllanthus emblica* | Leaf |
| 11 | AL2 | | Leaf |
| 12 | AS1 | | Stem |
| 13 | AS2 | | Stem |
| 14 | PL1 | *Mentha spicata* | Leaf |
| 15 | PL2 | | Leaf |
| 16 | PS1 | | Stem |

Voges-Proskauer (VP), urease production, and mannitol salt fermentation, isolates were biochemically analyzed [42]. Then, the standard protocol of Bergey's Manual of Systemic Bacteriology was followed for identification of isolates provisionally up to the species level [43].

### Determination of antibiotic sensitivity

The susceptibility of five finally identified isolates to different antibacterial agents was measured *in vitro* by employing the modified Kirby-Bauer method [44]. This method helps to determine the efficiency of a drug rapidly by measuring the diameter of the zone of inhibition that results from diffusion of the agent into the medium surrounding the disc [45]. Ten commercially available antibiotic discs (Himedia, India) were used for the tests (**Table 4**).

### Statistical analysis

Triplicate data were taken in all the cases during isolation, biochemical analysis and antibiotic sensitivity tests of the selected isolates. The results were analyzed according to the mean value ± standard deviation (SD) in triplicate. Microsoft Excel Software, version 2010, was used to calculate means and standard deviations by capturing all relevant data.

### Molecular identification of bacteria

Genomic DNA was extracted [46] and stored at -20˚C. Following a standard protocol, the DNA concentration was measured by a Thermo Scientific NanoDrop 2000 spectrophotometer (Thermo Scientific, USA) following a standard protocol. Polymerase chain reaction (PCR) was carried out for the detection of bacteria using previously published primers and targeted genes [47, 48]. The Basic Local Alignment Search Tool (BLAST) was used to determine primer specificity by searching for similar sequences in the microbial genome. During all experiments, positive and negative controls were carried out. The total composition of the target gene, primer sequences, cycling parameters, PCR master mixture and amplicon size (bp) were determined for PCR amplifications in a thermal cycler (NyxTechnik) and are shown in **Table 5**.

Amplified PCR products were then analyzed by electrophoresis (Micro-Bio-Tech Brand) in 2% (w/v) agarose gel in 1×TAE buffer, stained with ethidium bromide (1%) and compared with marker DNA (GeneRuler 1 kb DNA Ladder), finally visualized under ultraviolet (UV)

**Table 4. Antibiotic sensitivity test result.**

| Serial No. | Name of Antibiotics | Disc Code | Disc Potency (µg) | Diameter of Inhibition (mm) | | | | |
|---|---|---|---|---|---|---|---|---|
| | | | | LL1 | LL2 | LF | GL | GR |
| 1 | Penicillin G | P | 10 | 0 (R) | 20 (R) | 21(R) | 13 (R) | 20 (R) |
| 2 | Chloramphenicol | C | 30 | 29 (S) | 24 (S) | 30 (S) | 27 (S) | 30 (S) |
| 3 | Ampicillin | AMP | 25 | 12 (I) | 25 (S) | 23 (S) | 20 (S) | 24 (S) |
| 4 | Streptomycin | S | 10 | 23 (S) | 33 (S) | 24 (S) | 24 (S) | 25 (S) |
| 5 | Vancomycin | VA | 30 | 23 (S) | 30 (S) | 25 (S) | 25 (S) | 25 (S) |
| 6 | Norfloxacin | NX | 10 | 25 (S) | 21 (S) | 30 (S) | 30 (S) | 32 (S) |
| 7 | Tetracycline | TE | 30 | 22 (S) | 12 (R) | 24 (S) | 20 (S) | 22 (S) |
| 8 | Nalidixic Acid | NA | 30 | 26 (S) | 26 (S) | 29 (S) | 27 (S) | 30 (S) |
| 9 | Erythromycin | E | 15 | 27 (S) | 13 (R) | 28 (S) | 25 (S) | 29 (S) |
| 10 | Ceftriaxone | CTR | 30 | 26 (S) | 15 (R) | 31 (S) | 22 (S) | 32 (S) |

**Note**: R = Resistant, S = Sensitive, I = Intermediate

**Table 5. Target genes, primer sequences, cyclic conditions, PCR master mixture composition and amplicon size.**

| Target gene | Primer sequence (5´-3´) | Cycling parameters | Total Composition of PCR master mixture | Amplicon size (bp) | Reference (s) |
|---|---|---|---|---|---|
| Common Bacterial 16S rDNA | 8F-AGAGTTTGATCCTGGCTCAG 805R-GACTACCAGGGTATCTAAT | 5 min at 95°C, 35 cycles of 95°C for 40 s, 57°C for 50 s and 72°C for 1 min | For 10 µl: 5 µl master mix, 2 µl template, 1 µl[a] and 1 µl[b], 1 µl water | 800 | [47, 48] |

a = forward primer; b = reverse primer; s = seconds

trans-illuminator (Benda company) and then photographed. Then, PCR products were purified by an ATP™ Gel/PCR Fragment DNA Extraction Kit (Catalog No. ADF100/ADF300).

Five biochemically identified bacterial isolates were then sent for sequencing (Macrogen, South Korea). After sequencing, the results were visualized in DNA Baser software (V 5.15) and analyzed by the BLAST program in NCBI [49]. Then, the sequences were submitted to the Gen-Bank database. Phylogenetic tree construction and evolutionary analyses were performed for every five isolates between BLAST searches of identified bacteria using MEGA 11 software [50]. The maximum composite likelihood method was used for computing evolutionary distances [51].

### Screening for growth-promoting parameters

**Indole acetic acid (IAA) production.** The IAA production potential was calculated as per Gordon and Weber [46]. The endophytic bacterial isolates were grown on ISP2 broth containing 0.2% L-tryptophan incubated at 37°C with shaking at 150 rpm for 5 days. Cultures were centrifuged at 12,000 rpm for 10 min. Development of a pink–red color confirms IAA production by the addition of 0.5% Salkowski reagent into 1 ml of cell free supernatant. Estimation of IAA was measured by taking the absorbance at 530 nm using a spectrophotometer, and the amount of IAA was calculated in µg/ml compared with the standard curve of IAA.

**Ammonia production.** Ammonia production was estimated using the modified qualitative method [6], where endophytic isolates were incubated in peptone water broth at 37°C at 150 rpm for 7 to 14 days. Then, 0.5 ml of Nesseler's reagent was added to water broth, and the development of a brown to yellow color confirmed ammonia production. The absorbance was measured at 530 nm using a spectrophotometer, and ammonia production was expressed in mg/ml when compared with the standard curve of $(NH_4)_2SO_4$.

**Production of hydrolytic enzymes.** The proteolytic activity of endophytic bacteria was determined by streaking the isolates on skim milk agar medium. The isolates were single-streaked in skim milk agar medium and incubated at 37°C for 24–48 hours. A clear hollow zone around the areas where the organism has grown indicates proteolytic activity.

**Preparation of crude extract.** Isolated endophytic bacterial crude extracts were prepared following the methods described by Deljou and Goudarzi, 2016 with some modifications [52]. Endophytic bacterial isolates were inoculated in a 125 mL Erlenmeyer flask containing 25 mL nutrient broth. A rotary incubator shaker was used for incubation at 150 rpm and 37°C for 24 hours and 48 hours. After incubation, centrifugation was performed at 12,000 rpm for 10 minutes, and the cell and supernatant were extracted with organic solvent (1:1 v/v) using ethyl acetate (EA). A rotary vacuum evaporator was used to retrieve crude extracts by evaporating the organic solvents. The dry weight of the crude extracts was measured using a digital weighing machine and dissolved in 1% dimethyl sulfoxide (DMSO).

### Determination of antioxidant compounds

**Total phenolic content (TPC).** The total phenolic content was measured by following Folin-Ciocalteu's colorimetric method [53], where 0.1 mL of sample and 0.5 mL of Folin-

Ciocalteu were mixed with 6.0 mL of double-distilled water. After 1 min, 1.5 mL of 20% $Na_2CO_3$ (Merck, Germany) was added, and the total volume was made up to 10.0 mL with double-distilled water. The mixture was incubated for 2 h at 25°C. The absorbance was measured at 760 nm using a spectrophotometer against the blank solution containing all the reagents and the appropriate volume of the same solvent used for the sample. Gallic acid was used as the control indicator containing all the reaction agents except the sample.

**Total flavonoid content (TFC).** The total flavonoid content was measured by using the $AlCl_3$ colorimetric method [54] with slight modifications. Quercetin was used to make the calibration curve. One milligram of quercetin was dissolved in methanol and then diluted to 20, 40, 60, 80, and 100 µg/mL. Then, 0.5 mL diluted standard solutions were separately mixed with 1.5 mL of methanol, 0.1 mL of 10% $AlCl_3$, 0.1 mL of 1 M potassium acetate, and 2.8 mL of distilled water. After incubation at room temperature for 30 min, the absorbance of the reaction mixture was measured at 415 nm using a UV–Vis spectrophotometer. The amount of 10% $AlCl_3$ was substituted by the same amount of distilled water in the blank. Similarly, 0.5 mL of sample solution was used with $AlCl_3$ for determination of flavonoid content as described above.

## *In silico* analysis of bioactive secondary metabolite genes and compounds

**Detection of bioactive secondary metabolite regions.** The antiSMASH (antibiotics and Secondary Metabolite Analysis Shell) webserver tool (bacterial version) was used to detect the secondary metabolite regions of the isolated endophytes. First, the whole genome sequences and accession numbers of the bacterial strains were obtained from NCBI, and then sequences were uploaded as data input. KnownClusterBlast, SubClusterBlast, and ActiveSiteFinder were marked as featured options. From the graphical output, secondary metabolite regions and their types were identified.

**Prediction of bioactive secondary compounds.** The antiSMASH (antismash. secondarymetabolites.org) and Minimal Information about a Biosynthetic Gene cluster-MIBiG (mibig.secondarymetabolites.org) databases were used to identify the proximate secondary metabolite regions. According to the cluster, different compounds were found as hits through Knowledge Blast.

**Analysis of biosynthetic genes and interaction of proteins.** Secondary metabolite gene clusters were searched for core and additional biosynthetic genes involved in metabolite production. Later, the STRING database (string-db.org) was used to identify and analyze the protein–protein interaction between the core genes involved in secondary metabolism.

**Analysis of PKS/NRPS domains.** Statistical preferences of putative polyketide synthase (PKS) and nonribosomal peptide synthetase (NRPS) domains at the genus level were analyzed via the SBSPKS v2 web server tool (www.nii.ac.in/~pksdb/sbspks) [55]. Datasets were generated from the MIBiG repository.

**Molecular detection of biosynthetic genes.** Four sets of degenerate primers were designed according to the study Ayuso-Sacido and Genilloud, 2005 [56], targeting the universal bacterial PKS (polyketide synthase) and NRPS (nonribosomal peptide synthetase) genes along with chalcone synthase (CHS) and ACC deaminase (acdS) gene primers.

PCR conditions were as follows: initial denaturation at 95°C for 4 min, followed by 30 cycles of denaturation at 94°C for 1 min, annealing at 57°C for NRPS, 58°C for PKS, 52°C for CHS and 55°C for ACCD primers for 1 min and extension at 72°C for 1 min with a final extension step at 72°C for 4 min.

**Molecular docking of bioactive compounds against the SARS-COV2 protein.** The crystal structures of coronavirus 3CL^pro used in the docking analysis were retrieved from the

Protein Databank (http://www.rcsb.org) with its PDB identification code (6lu7) [57, 58]. The 3CL$^{pro}$ structures were processed by eliminating existing ligands and water molecules, while missing hydrogen atoms were added according to the amino acid protonation state at pH 7.0 employing the Autodock version 4.2 program (Scripps Research Institute, La Jolla, CA) [59].

The structure data format (SDF) structures of the 3CL$^{pro}$ reference inhibitor lopinavir and the identified endophytic bioactive compounds were retrieved from the PubChem database (www.pubchem.ncbi.nlm.nih.gov) [60].

Virtual screening of the coronavirus 3CL$^{pro}$ active regions and determination of the binding affinities of the compounds and reference inhibitor were carried out using AutoDockvina 4.2 with full ligand flexibility. The molecular interactions between proteins and respective ligands with higher binding affinity were viewed using Discovery Studio Visualizer version 16.

**ADMET and drug-likeness prediction.** After the molecular docking studies, the absorption, distribution, metabolism, elimination and toxicity (ADMET) of the docked metabolites were screened using the online tools admetSAR (lmmd.ecust.edu.cn) [61] and SwissADME (www.Swissadme.ch) to predict their important pharmacokinetic properties [62].

**1H-NMR spectrophotometry.** The bacterial crude extract was sent to INARS Laboratory, Bangladesh Council of Scientific and Industrial Research (BCSIR), DHAKA for 1HNMR analysis. At BCSIR, data were acquired on a 400 MHz spectrometer (Bruker Corporation).

## Results

### Isolation of endophytic bacteria

Forty-eight plant parts (3 parts from 16 plants each) were subjected to surface sterilization [63], triturated with autoclaved phosphate-buffered saline (PBS) [64, 65], and then used to isolate endophytic bacteria. After spreading the plant extracts derived from forty-eight (48) plant parts over solid nutrient agar (NA) media, several bacterial colonies were found, and sixteen pure colonies were selected (**Table 3**) after incubation for 48 hours at 37˚C. Then, several subcultures were performed to isolate pure cultures from different types of endophytic bacteria (**Fig 1**) and finally preserved in a 4˚C refrigerator for further studies.

**Identification of endophytic bacteria by biochemical characterization.** Pure cultures of five selected types of isolates were then subjected to a series of biochemical tests (**Table 6**) as described in Bergey's Manual for Determinative Bacteriology [66]. After the Gram staining procedure, four isolates, LL1, LL2, LF, and GL, showed violet colors, whereas GR showed pink colors under a microscope. Thus, GR was considered gram-negative, and the others were considered gram-positive bacteria. LL1, LF, and GR were rod in shape under a microscope, while LL2 and GL were identified as coccus-shaped bacteria. In the catalase test, four samples, LL1, LL2, LF, and GL, produced bubbles after the addition of $H_2O_2$, which indicated the presence of the catalase enzyme. Sample GR failed to produce any bubbles and thus was catalase-negative. In the oxidase test, all five isolates, LL1, LL2, LF, GL, and GR, showed a deep blue color within 10 seconds, indicating the ability to produce oxidase enzymes as a positive result. In the indole production test, the appearance of a red ring indicates a positive result. All five isolates tested negative, as no red ring appeared, indicating the absence of tryptophanase enzyme. Four isolates, LL1, LL2, LF, and GL, exhibited red color formation, indicating the capacity to produce and maintain a stable acid end product. Sample GR, on the other hand, was unable to show red color formation and thus tested negative. In the Voges-Proskauer test, all five samples were unable to yield red color formation, which indicated that they were unable to produce butylene glycol. Thus, all the samples were tested negative. In the citrate utilization test, only isolate LL2 was found to produce a blue color as a positive result, which indicated that the other four isolates were unable to utilize citrate due to the absence of citrate-utilizing enzymes.

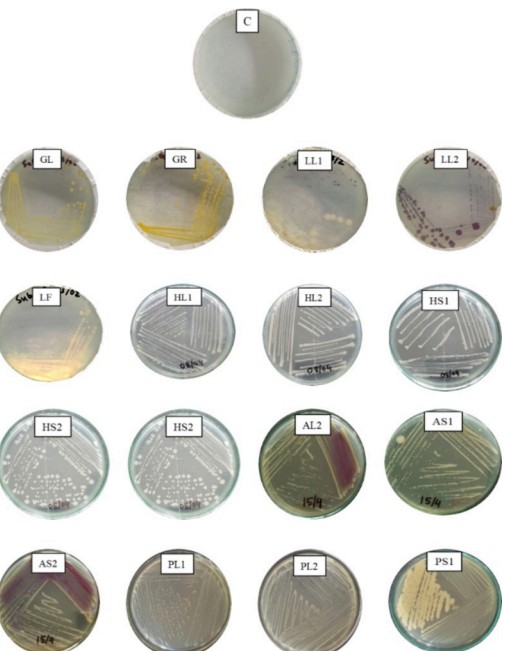

**Fig 1. Pure cultures of sixteen isolated endophytic bacteria labeled as their sample codes (GL, GR, LL1, LL2 LF, HL1, HL2, HS1, HS2, AL1, AL2, AS1, AS2, PL1, PL2, PS1); Control (C) indicates no growth of bacteria in the medium.**

Mannitol salt agar (MSA) media was used to identify the isolates growing on high salt concentrations, and a positive test consisted of a color change from red to yellow, indicating a pH change to acidic. Among the five isolates, only LL1 showed a positive result (**Table 6**).

**Antibiotic sensitivity test.** Ten types of antibiotic discs (Himedia, India) were used to test the sensitivity [44], where every isolate showed resistance toward penicillin G. In addition, isolate LL2 showed resistance (R) to tetracycline, erythromycin and ceftriaxone. LL1 expressed

**Table 6. Summary of morphological and biochemical characterization.**

| Morphological and Biochemical Characteristics | ISOLATE CODE | | | | |
|---|---|---|---|---|---|
| | **LL1** | **LL2** | **LF** | **GL** | **GR** |
| *Morphological Characteristics* | | | | | |
| Colony Color | Yellowish | Purple | White | Yellowish | Yellow |
| Gram Staining | + | + | + | + | - |
| Shape | Rod | Cocci | Rod | Cocci | Rod |
| *Biochemical tests* | | | | | |
| Catalase | + | + | + | + | - |
| Oxidase | + | + | + | + | + |
| Indole | - | - | - | - | - |
| Methyl Red | + | + | + | + | - |
| VP | - | - | - | - | - |
| Mannitol Salt Fermentation | - | + | - | - | - |
| Citrate Utilization | + | - | - | - | - |
| **Identified Strain** (provisionally) | *Bacillus spp.* | *Staphylococcus spp.* | *Bacillus spp.* | *Staphylococcus intermedius* | *Pseudomonas spp.* |

Note: (+) indicates a positive result and (-) indicates a negative result.

intermediate (I) sensitivity to ampicillin [67], and the rest of the isolates were found to be sensitive to all antibiotics (**Table 4**).

**Identification of bacteria depends on morphological and biochemical analysis.** The overall morphological and biochemical test results of five distinct isolates and their identifications [66] are listed in **Table 6**.

**Identification of bacterial strains by molecular characterization.** After extraction of genomic DNA from five isolates, DNA concentration and purity were measured by a Nanodrop 2000 (Thermo Scientific, USA). Extracted DNA of five bacterial isolates was then amplified by 16S rDNA primers. Gel electrophoresis was performed afterwards on a 2% agarose gel and stained with ethidium bromide. After that, the gel was visualized by a UV transilluminator (Benda Company) (**Fig 2**). After PCR amplification of five bacterial isolates, they were sent for Sanger sequencing by Macrogen's sequencing service (South Korea). Purified PCR products along with their respective primers were sequenced and finally confirmed up to their species. Isolates LL1, LL2, LF, GL and GR were thus identified as *Priestia megaterium*, *Staphylococcus caprae*, *Neobacillus drentensis*, *Micrococcus yunnanensis*, and *Sphingomonas paucimobiliz*, respectively, with 99% identity with an e value of 0 in NCBI BLAST (Basic Local Alignment Search Tool) analysis. After analysis by the BLAST program, isolate information was submitted to GenBank by using the GenBank submission portal. Then, the accession numbers for the nucleotide sequences were obtained as MW494406, MW494408, MW494401, MW494402, and MZ021340.

The five isolates were then subjected to phylogenetic tree construction according to their 16S rRNA sequences with MEGA 11 Version 5.0 Software (**Fig 3**). The neighbor-joining method was used to interpret evolutionary history. The optimal tree is demonstrated with the sum of branch length = 0.06596246. The scale is shown in the tree, with branch lengths in the same units as those of the evolutionary distances used to infer the phylogenetic tree. The evolutionary distances were enumerated by applying the maximum composite likelihood method [51] in units of the number of base substitutions per site. The analysis has 20 nucleotide sequences. Codon positions comprised with 1st+2nd+3rd+Noncoding. All cryptic positions were eliminated for each sequence pair. There were a total of 1349 positions in the final dataset. Evolutionary analyses were conducted in MEGA 11 [50].

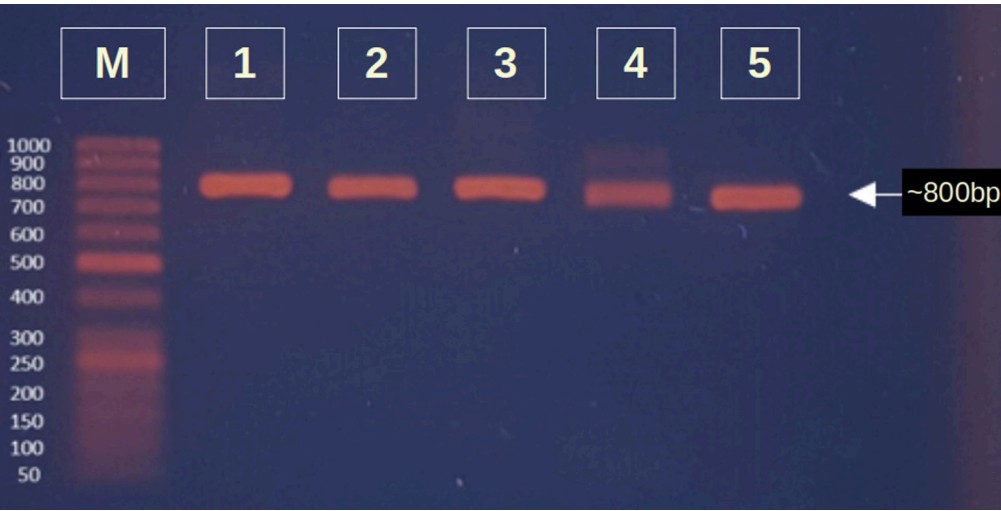

**Fig 2. Electrophoretic separation (2% agarose) of the 16S rDNA gene of different isolates. M:** 50 bp DNA ladder; **1:** LL1; **2:** LL2; **3:** LF; **4:** GL; **5:** GR.

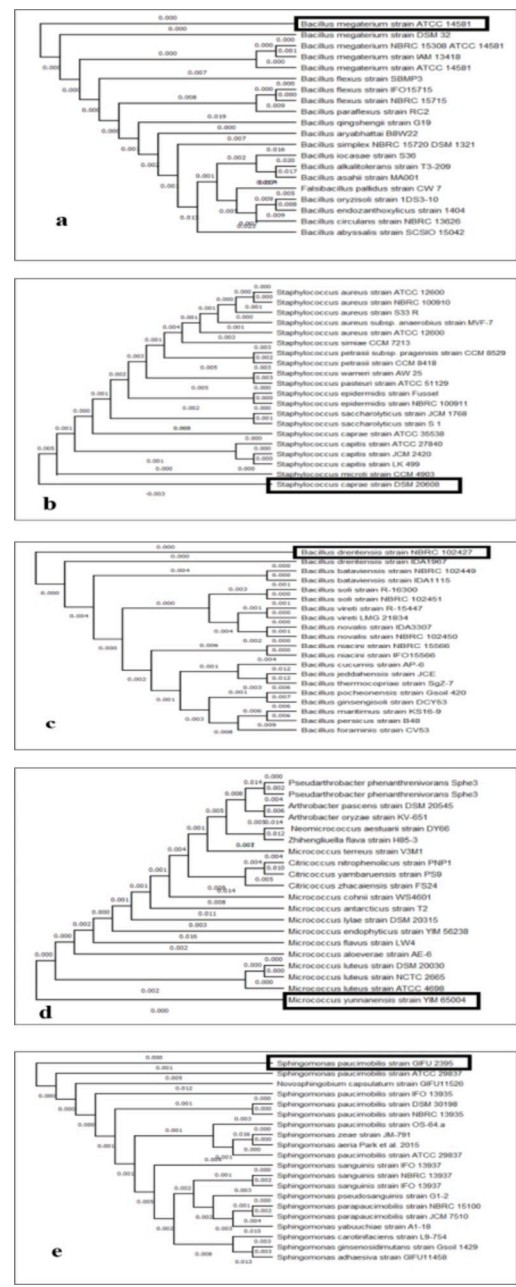

**Fig 3. Phylogenetic tree of isolates.** (a) *Priestia megaterium*, (b) *Staphylococcus caprae*, (c) *Bacillus drentensis*, (d) *Micrococcus yunnanensis*, (e) *Sphingomonas paucimobiliz*.

**Screening for growth-promoting parameters of isolated bacteria.** Quantitative estimation of all isolates was performed in ISP2 broth medium [68], and the indole acetic acid (IAA) production rate ranged from 14.26 to 25.56 µg/mL (**Table 7**). A maximum IAA yield of 25.561 ± 0.05 µg/mL was observed by isolate GR (*Sphingomonas paucimobiliz*). IAA solution was used as a standard, and the concentration of samples was estimated by the standard curve (**Fig 4**).

Quantitative estimation of ammonia production by all isolates in peptone water broth ranging from 5.3 to 24.98 mg/mL. Isolate LL2 (*Staphylococcus caprae)* produced the maximum

**Table 7. Estimation of IAA, ammonia, TPC and TFC by five isolates.**

| Name of isolates | IAA | | Ammonia | | TPC | | TFC | |
|---|---|---|---|---|---|---|---|---|
| | Absorbance (530 nm) | Concentration (µg/mL) | Absorbance (530 nm) | Concentration (µg/mL) | Absorbance (760 nm) | Concentration (µg/mL) | Absorbance (415 nm) | Concentration (µg/mL) |
| *Priestia megaterium* (LL1) | 0.416 | 14.268 ± 0.05 | 1.074 | 20.23 ± 0.01 | 0.327 | 258.901 ± 0.02 | 0.2237 | 36.09 ± 0.02 |
| *Staphylococcus caprae* (LL2) | 0.727 | *23.902 ± 0.06 | 1.211 | **24.98 ± 0.03 | 0.467 | **406.653 ± 0.01 | 0.2971 | **45.18 ± 0.06 |
| *Neobacillus drentensis* (LF) | 0.646 | *23.612 ± 0.01 | 0.662 | 5.98 ± 0.01 | 0.366 | 300.060 ± 0.01 | 0.2381 | 38.92 ± 0.05 |
| *Micrococcus yunnanensis* (GL) | 0.536 | 19.143 ± 0.20 | 0.615 | 4.35 ± 0.02 | 0.314 | 245.181 ± 0.05 | 0.1380 | 19.29 ± 0.01 |
| *Sphingomonas paucimobiliz* (GR) | 0.694 | **25.561 ± 0.05 | 0.644 | 5.32 ± 0.04 | 0.302 | 232.517 ± 0.05 | 0.1416 | 20.01 ± 0.02 |

Note: ** indicates maximum result

amount of ammonia (24.98±0.03 mg/mL) (Table 7). Ammonium sulfate was used as the standard, and the concentration of samples was estimated by the standard curve (**Fig 4**).

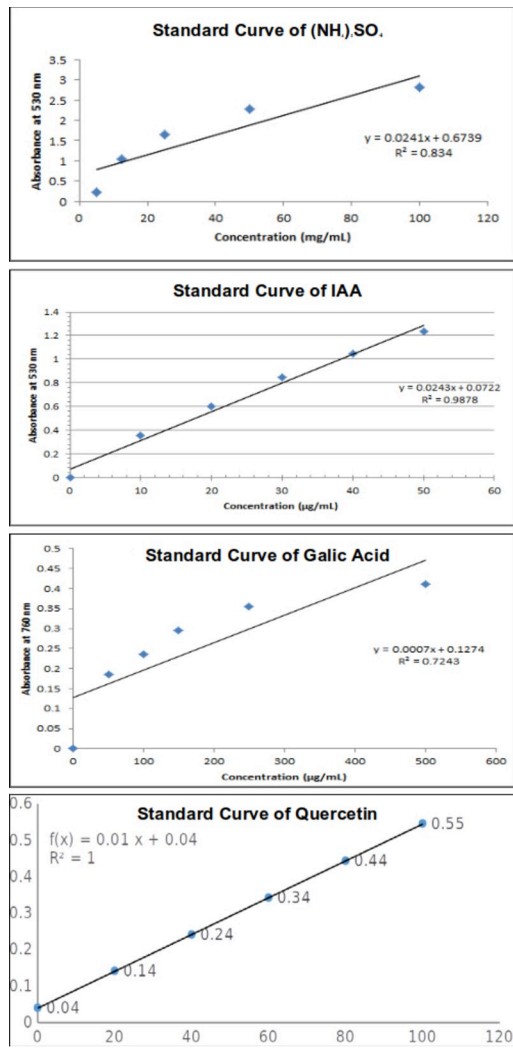

**Fig 4. Standard curves of ammonium sulfate, indole acetic acid, gallic acid and quercetin.**

Out of five isolates, two isolates were found to be positive for protease enzyme activity. *Priestia megaterium* (LL1) and *Staphylococcus caprae (*LL2) isolates showed clear degradation of the protein (**Fig 5**) in skim milk agar [69].

**Determination of antioxidant compounds from crude extracts.** Crude extracts (extracellular secondary metabolites) of isolated endophytic bacteria were estimated by using ethyl acetate solvent, and each sample was incubated for 24 hours and 48 hours [52]. *Staphylococcus caprae* (LL2) showed the maximum result, while *Sphingomonas paucimobiliz* (GR) showed the minimum (**Table 8**). Ethyl acetate (EA) extracts were stored in 1% dimethyl sulfoxide (DMSO) for further antioxidant and metabolite analysis [70].

Ethyl acetate extracts of the isolated bacteria were processed to analyze total phenolic compounds [71]. Isolate LL2 exhibited the maximum result (406.653±0.01 μg/mL), and isolate GR exhibited the lowest (232.517±0.05 μg/mL) (**Table 7**). Gallic acid was used as the standard to estimate the TPC (**Fig 4**).

Ethyl acetate extracts of the isolated bacteria were processed to analyze total flavonoid compounds. Isolate *Staphylococcus caprae* (LL2) exhibited the maximum result (45.18±0.06 μg/mL), and isolate *Micrococcus yunnanensis* (GL) exhibited the lowest result (19.29±0.01 μg/mL) (**Table 7**). Quercetin was used as the standard [70] to estimate the TPC (**Fig 4**).

***In silico* analysis of bioactive secondary metabolite genes and compounds.** Antibiotics and secondary metabolite analysis shell (antiSMASH) server was used to identify different secondary metabolite regions present in the genome of identified bacterial strains (**Fig 6**). The whole-genome sequences were taken from NCBI and input to the antiSMASH server. A total of 13 different bioactive secondary metabolite regions were identified within five isolates. Terpene was the most common type of cluster found in each isolate, followed by Type III polyketide synthase (T3PKS) and the siderophore cluster. *Priestia megaterium* had the highest number (9) of bioactive regions among the isolates (**Fig 7**).

After the identification of core biosynthetic gene clusters by antiSMASH, the MIBig (Minimal Information about a Biosynthetic Gene cluster) specification was used to predict the bioactive compounds according to the region and their type (**Table 9**). Nine (9) putative compounds were found from the whole genome sequences of identified endophytes. The most similar known cluster was searched for the identified query clusters, and based on a percentage

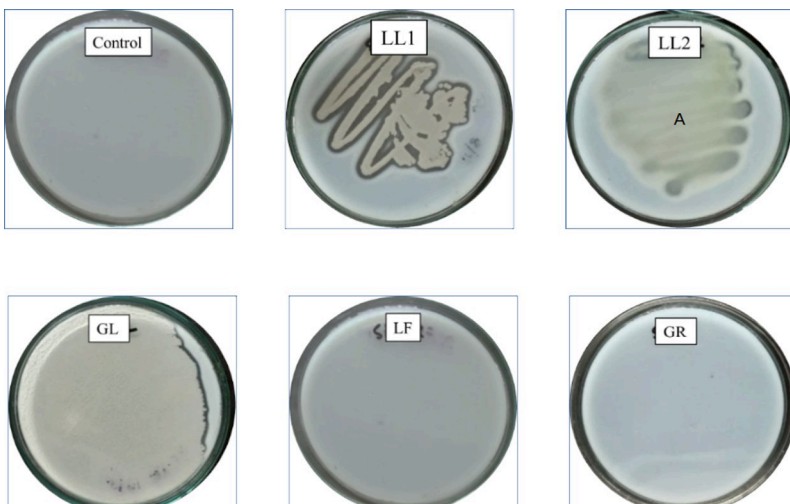

**Fig 5. Isolates in skim milk agar.** LL1 (Priestia megaterium) and LL2 (Staphylococcus caprae) represented protease enzyme activity with a clear hollow degradation of protein throughout the inoculation area.

**Table 8. Production of crude extracts from isolated endophytic bacteria.**

| Name of isolates | Incubation Period (hour) | Quantity of Crude Extract (gm) |
|---|---|---|
| *Priestia megaterium* (LL1) | 24 | 0.34 ± 0.02 |
| | 48 | 0.36 ± 0.03 |
| *Staphylococcus caprae* (LL2) | 24 | **0.45 ± 0.01 |
| | 48 | **0.48 ± 0.01 |
| *Neobacillus drentensis* (LF) | 24 | 0.16 ± 0.02 |
| | 48 | 0.22 ± 0.03 |
| *Micrococcus yunnanensis* (GL) | 24 | 0.14 ± 0.04 |
| | 48 | 0.21 ± 0.01 |
| *Sphingomonas paucimobiliz* (GR) | 24 | 0.12 ± 0.05 |
| | 48 | 0.17 ± 0.04 |

**Note**: ** indicates maximum results

similarity, their molecular compounds were predicted as hits (**Fig 8**). MIBiG produced the output file with details of biosynthetic genes (core and additional), transport-related genes, regulatory genes and other genes. Compound structures were also obtained from the output (**Fig 8**). Predicted similar compounds (**Table 9**) were analyzed through their core and additional biosynthetic genes obtained from the antiSMASH and MIBiG datasets (**Table 10**). The interaction of those genes and their protein products was checked through the STRING database (version 11.0) [72]. All genes associated with those 9 gene clusters of identified compounds were programmed to be enlisted according to the query in the Uniprot database of protein. The highest 12 biosynthetic genes were identified in the gene cluster responsible for the fengycin compound and likely 9 genes for surfactin, 6 genes for bacitracin, 3 genes for zeaxanthin, and 4 genes for both the carotenoid and staphyloferrin A compounds (**Table 9**). All the genes were retrieved with enough protein data from the Uniprot server [73], but unfortunately, no genes could be retrieved for aureusmin, stenothrin, or microanamycin. As we have used the STRING server to expose the interaction between the genes and the proteins, these three compounds may be unable to produce any associated network. Interactions between genes involved in secondary

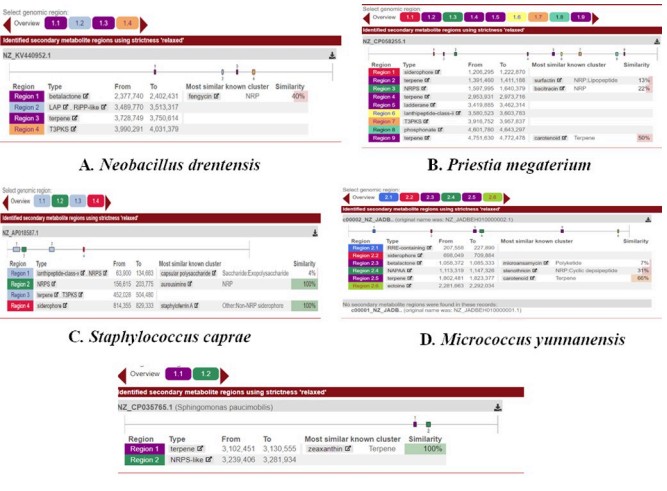

A. *Neobacillus drentensis*

B. *Priestia megaterium*

C. *Staphylococcus caprae*

D. *Micrococcus yunnanensis*

E. *Sphingomonas paucimobilis*

**Fig 6. Identification of secondary metabolite regions using the antiSMASH server.**

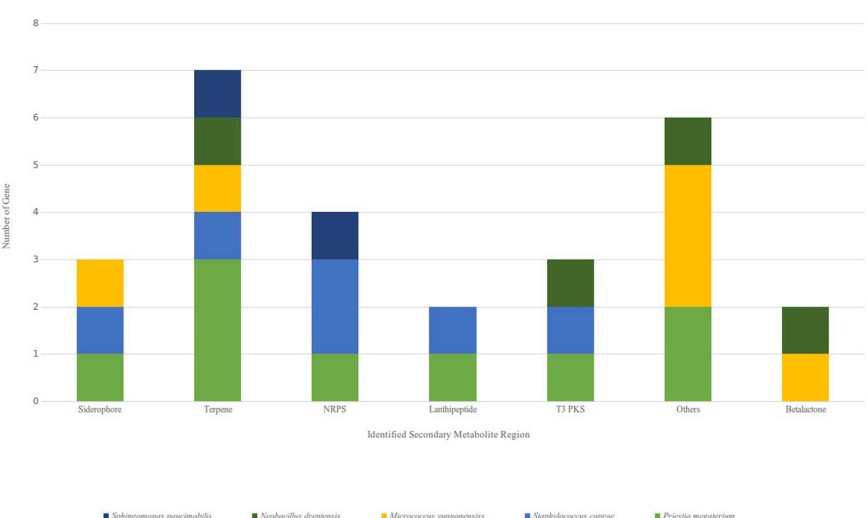

**Fig 7. Distribution of genes in identified biosynthetic gene clusters: NRPS Non ribosomal peptide synthetase, T3PKS = Type 3 polyketide synthase.**

metabolite gene clusters of the compounds are shown in nodes and edges (**Fig 9**), where significant protein–protein interactions were observed according to the functional properties of the genes and their respective products. ClusterBlast results (from the MIBiG database) were further analyzed to measure the PKS (polyketide synthase) and NRPS (nonribosomal peptide synthetase) in a comparative source of endophytic bacteria at their sequenced species level [19]. SBSPKS v2 was used to carry out functional analyses of the PKS and NRPS domains (**Fig 10**). *Bacillus* was found to be rich in these domains, having confirmed and some hypothetical regions. A moderate percentage was found for *Staphylococcus* spp. possessing similar amounts and thus exhibited diverse types of PKSs and NRPSs. These particular conserved clusters are spotted domains of Fimicutes, Proteobacteria and Actinobacteria phyla [15].

The presence of biosynthetic genes was determined in the identified strains with four sets of gene-specific primers. PCR amplification of polyketide synthase (PKS), nonribosomal peptide synthetase (NRPS) [56], 1-aminocyclopropane-1-carboxylate deaminase (ACCD) [4], and chalcone synthase (CHS) [19] genes was performed for our identified strains. All strains showed a prominent band (amplicon size ~600–700 bp) for the NRPS gene [83] (**Fig 11B**). PKS candidate amplicons (~700–800 bp) were detected in *Priestia megaterium* (LL1), *Staphylococcus caprae* (LL2), *Micrococcus yunnanensis* (GL), and *Sphingomonas paucimobiliz* (GR)

**Table 9. List of predicted bioactive compounds from gene clusters.** \*\* All the structures of the compound are given in the (**S1 Fig**).

| Identified isolates | Compound | Type | Bioactivity |
|---|---|---|---|
| *Priestia megaterium* (LL1) | Surfactin | Nonribosomal peptide (NRP)-lipopeptide | Surfactant, Antibacterial Antiviral [74] |
| | Bacitracin | Nonribosomal peptide (NRP) | Broad-spectrum antibiotic [75] |
| | Carotenoid | Terpene | Antioxidant, a precursor of vitamin [76] |
| *Micrococcus yunnanensis* (GL) | Microansamycin | NRP- Cyclic depsipeptide | Antioxidant [77] |
| | Stenothricin | Terpene (nonalpha poly-amino acids like e-Polylysin) | Antibiotic [78] |
| *Staphylococcus caprae* (LL2) | Aureusimine | Nonribosomal peptide (NRP) | Potent antibiotic [79] |
| | Staphyloferrin A | Siderophore | Potent antibiotic [80] |
| *Neobacillus drentensis* (LF) | Fengycin | Betalectone | Antifungal [81] |
| *Sphingomonas paucimobiliz* (GR) | Zeaxanthin | Terpene | Antioxidant [82] |

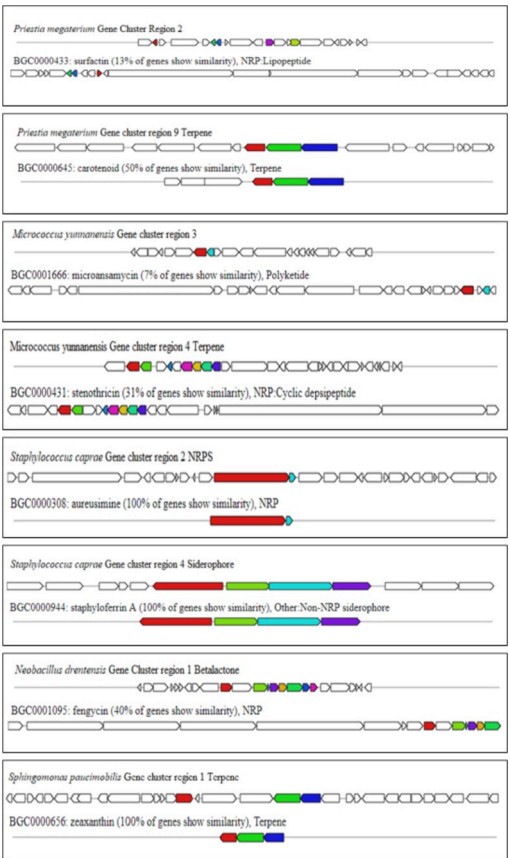

**Fig 8. Graphical representations of most similar known clusters as an output of ClusterBlast specified by the MIBiG repository.** Color codes represent similar gene regions.

(**Fig 11A**). For the ACCD gene, only *Sphingomonas paucimobiliz* (GR) was positive (**Fig 11C**), and all isolates showed negative results for the CHS gene, as no DNA band was found in PCR amplification (**Fig 11D**).

The binding energy from the docking analysis of bioactive compounds (5) and reference compound to 3-chymotrypsin-like protease (3CL$^{pro}$) of the novel SARS-COV-2 is represented in **Table 11**. The ligand–protein binding interactions according to the residues of our compound and macromolecule (3CL$^{pro}$) are shown in **Fig 12A–12F**. Protein–ligand interactions were exhibited through residues and bonds via 2D representation in **Fig 13**.

The results generated from the Lipinski and ADME/tox filtering analyses are presented in **Table 12.** Two compounds were found to be **required** for Lipinski analysis of the rule of five with corresponding favorable predicted ADME/tox parameters. The predicted physiochemical properties for the bioavailability of the lead compounds are further represented in **Fig 14.**

**1H-NMR metabolomics analysis.** With respect to the 1H-NMR metabolite data, the proton number was calculated for each metabolite and we have found three metabolites: microansamycin (**Fig 15A**), aureusimine (**Fig 15B**) and surfactin (**Fig 15C**) in our bacterial sample.

## Discussion

Plants possess thousands of microbes residing inside their various tissues and forming a mutual relationship. Every living plant on earth is host to one or more endophytes: bacteria or fungi that colonize living plant tissues by producing a wide variety of specialized metabolites

**Table 10. List of responsible genes and proteins from the secondary metabolite gene clusters of the identified compounds.**

| Compound | Gene | Protein | Biological Process | Pathway |
|---|---|---|---|---|
| Surfactin | srfAA | Surfactin synthase subunit 1 | Antibiotic biosynthetic process | Surfactin biosynthesis |
| | srfAB | Surfactin synthase subunit 2 | | |
| | srfAC | Surfactin synthase subunit 3 | | |
| | srfAD | Surfactin synthase thioesterase subunit | Transport | Transmembrane transport |
| | ycxA | MFS type Transporter | Transport | Transmembrane transport |
| | ycxB | Uncategorized protein | | |
| | ycxC | Transporter Protein | Transport | Transmembrane transport |
| | ycxD | Transcriptional regulator protein | DNA binding cofactor | Alpha amino acid metabolic pathway |
| | tcyC | L-cystine import ATP-binding protein | ATPase-coupled amino acid transmembrane transporter activity | Amino acid transport pathway |
| Bacitracin | bacA | Prephenate decarboxylase | Antibiotic biosynthetic process | Bacilysin biosynthesis |
| | bacB | H2HPP isomerase | | |
| | bacC | bacilysin synthetase C | | |
| | bacT | Thioesterese family protein | | |
| | bacR | Transcriptional regulatory Protein | DNA binding | |
| | bacS | Histidine kinase | Regulation process | |
| Carotenoid | crtNa | CrtNa protein | Oxidoreductase activity | Carotenoid biosynthesis |
| | crtNc | CrtNc protein | oxidoreductase activity | |
| | crtM | 4,4'-diapophytoene synthase | Transferase activity | |
| | crtNb | CrtNb protein | Oxidoreductase activity | |
| Staphyloferrin A | sfaC (SAOUHSC_02433) | Staphyloferrin A synthetase | Nonribosomal peptide biosynthesis | Staphyloferrin biosynthesis |
| | sfaB (SAOUHSC_02434) | Staphyloferrin A synthetase | Nonribosomal peptide biosynthesis | |
| | sfaA (SAOUHSC_02435) | Staphyloferrin A Transporter | Siderophore transporter | |
| | iucC_3 (SAOUHSC_02436) | L-ornithine Racemase | Siderophore biosynthesis | Precursor biosynthesis |
| Zeaxanthine | crtY | Lycopene cyclase | lycopene beta cyclase activity | carotenoid biosynthetic process |
| | crtI | Phytoene dehydrogenase | oxidoreductase activity | carotenoid biosynthetic process |
| | crtB | Phytoene synthase | squalene synthase activity | |
| Fengycin | yngE | Acyl-CoA carboxylase subunit beta | ligase activity | |
| | yngF | Enoyl-CoA hydratase | catalytic activity | |
| | yngG | Hydroxymethylglutaryl-CoA lyase | oxo-acid-lyase activity | |
| | yngH | Acetyl-CoA carboxylase biotin carboxylase subunit | ATP binding | |
| | yngI | AMP-binding protein | AMP-binding | |
| | yngJ | Acyl-CoA dehydrogenase | Oxidoreductase | |
| | yngK | Glycoside hydrolase family protein | hydrolase activity | |
| | fenA | fengycin synthetase A | Nonribosomal peptide synthetase | Antibiotic biosynthetic process |
| | fenB | fengycin synthetase B | | |
| | fenC | fengycin synthetase C | | |
| | fend | fengycin synthetase D | | |
| | fenE | fengycin synthetase E | | |

without causing any harm or disease to the host plants [29]. However, in the case of medicinal plants, relatively few endophytes have been studied thus far, and of course, for the endemic

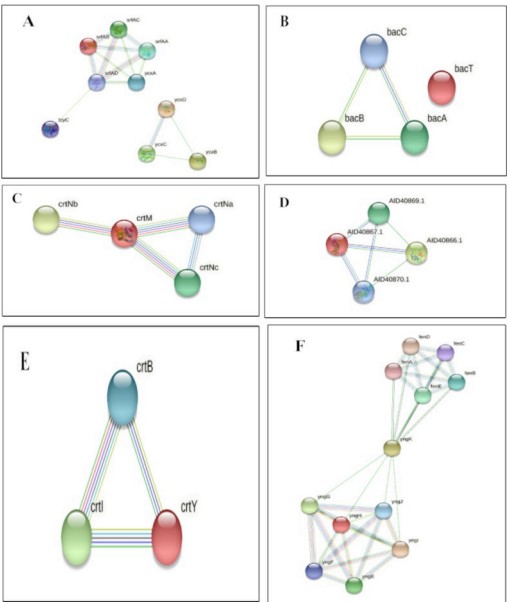

**Fig 9. Predicted association of proteins of biosynthetic genes demonstrated by the STRING server.** (A) Surfactin, (B) bacitracin, (C) carotenoid, (D) staphyloferrin A, (E) zeaxanthin and (F) fengycin protein network. Line Indicator: Red Line–Presence of fusion evidence, Green Line- neighborhood evidence, Blue Line- cooccurrence evidence, Purple Line- experimental evidence, Yellow Line- textmining evidence, Light Blue Line- database evidence, Black Line- coexpression evidence.

medicinal plants of Bangladesh as well. Therefore, one of the major objectives of this study is to isolate and characterize endophytic bacteria from our local medicinal plants to study their secondary metabolites, which have a wide range of applications in the global health sector.

Studies have revealed the capacity of endophytes to produce a diverse range of secondary metabolites [13]. Different types of important natural products, including antibiotics, antifungal, insecticidal, anticancer, immunosuppressant, antiviral, and volatile organic compounds, have been derived or synthesized from various endophytic bacteria. There has been growing

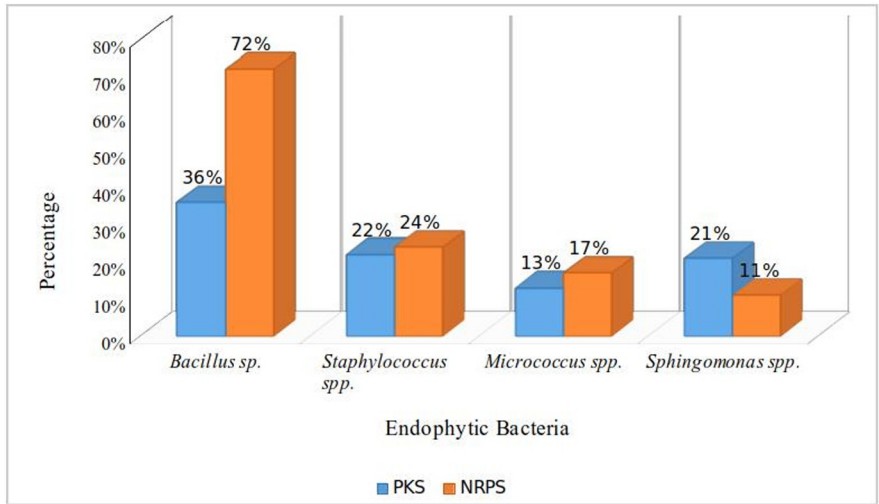

**Fig 10. Distribution of PKS and NRPS domains among the endophytic isolates at the genus level.**

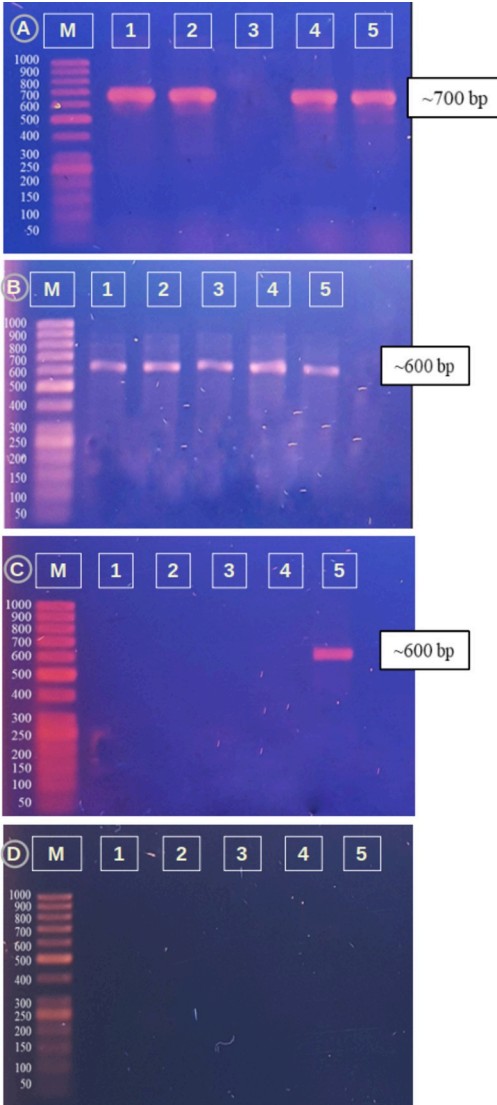

**Fig 11. PCR amplifications of biosynthetic genes.** (A) PKS gene, (B) NRPS gene, (C) ACCD gene and (D) CHS gene. **M:** DNA Marker for each case; **1:** LL1; **2:** LL2; **3:** LF; **4:** GL; **5:** GR.

**Table 11. Binding affinities of the reference compound (lopinavir) and other compounds to 3CL<sup>pro</sup> of coronaviruses.**

| Serial no. | Compound/Ligand | Binding Affinity (Kcal/mol) |
|---|---|---|
| 1. | Lopinavir (ref. compound) | -7.2 |
| 2. | Aureusimine | **-6.0** |
| 3. | Bacitracin | -3.8 |
| 4. | Carotenoid | -4.1 |
| 5. | Microansamycin | **-7.7** |
| 6. | Staphyloferrin | -5.8 |

**Note:** Compounds having the highest binding affinity for the corresponding proteins are the ones indicated in bold values.

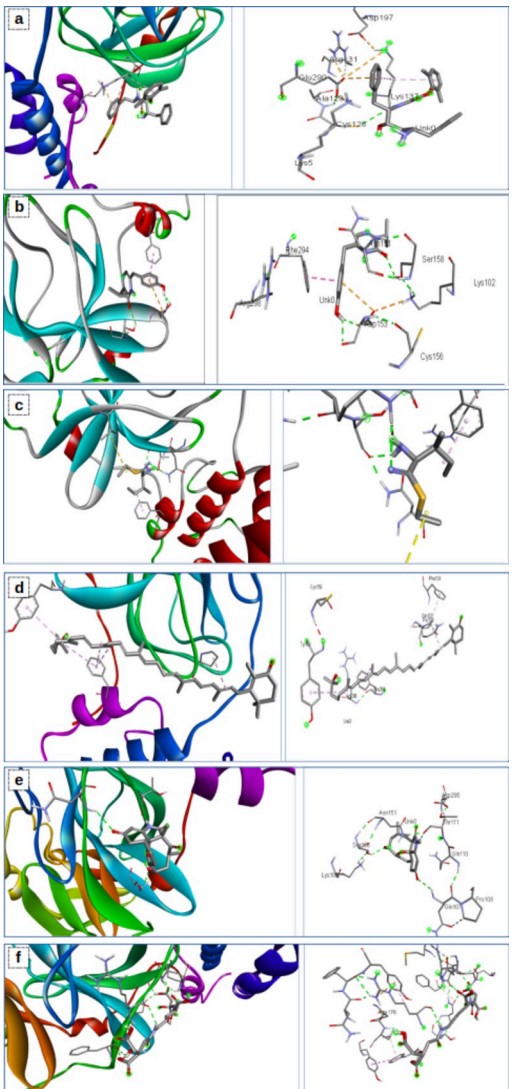

**Fig 12.** Visualization of SARS-CoV-2 3CLpro amino acid interactions with the ligands **(A)** lopinavir, **(B)** aureusimine, **(C)** bacitracin, **(D)** carotenoid, (E) microanasamycin, and (F) staphyloferrin.

interest in endophytic bacteria until last several years due to their efficacy in mimicking and producing similar bioactive compounds of their respective host plants as well as new bioactive compounds that are not present in host plants [9]. As medicinal plants harbor natural phytochemicals and bioactive compounds, endophytic bacteria within them can be a potential source of interest to investigate natural products [84]. Our study also aims to find a picture of bioactive potentiality inside endophytes by investigating biosynthetic gene clusters and secondary metabolites with molecular studies and *in silico* approaches.

In the subcontinent, approximately 2000 plants with medicinal properties have been reported, and among them, approximately 500 of such medicinal plants have thus far been enlisted as growing or available in Bangladesh [85]. In our study, we collected our plant samples around the campus of Chittagong University (**Table 2**), as it is a good reservoir of indigenous plants of Bangladesh [85]. As we are interested in identifying culturable endophytic bacteria from various parts of the plants, we collected root, stem, and leaf parts of each plant

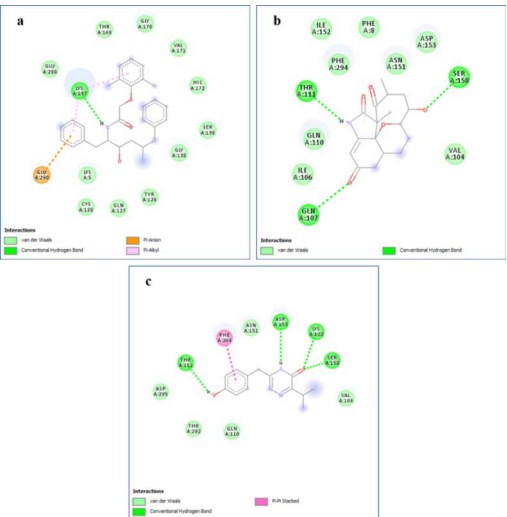

**Fig 13. 2D representation of protein–ligand interactions.** Binding mode of 3CLpro with A) lopinavir, B) microanasamycin, and C) aureusimine.

sample. We did not consider the flower and seed parts, as we selected different herbs, shrubs, and trees to avoid flowering and nonflowering properties [71]. In this study sixteen medicinal plant samples were collected from eight (8) different locations to obtain diversity.

The surface sterilization process was performed carefully to eliminate the epiphytes from the plant parts. Phosphate buffer solution (PBS) was used to triturate the internal plant extracts obtained from the leaves, roots, and stems [65]. The extracts were collected after trituration and then spread on the culture plates of nutrient agar (NA) media through the pour-plate technique [86]. We used nutrient agar (NA) and Luria-Bertani (LB) agar media to study further growth of isolated endophytic bacteria. Thus, 16 bacterial isolates (endophytes) were selected from forty eight medicinal plant parts which confirms the culture-dependent nature of those endophytic bacteria [5, 64]. Colonies were then separated by identifying colony color, shape and orientation for further pure culture (**Fig 1**) by the streaking method [29, 69]. Standard morphological and biochemical tests were performed to characterize and identify five bacterial isolates, LL1, LL2, LF, GL, and GR [66, 87]. In the case of gram staining study (**Table 6**), four isolates LL1, LL2, LF, and GL were considered gram-positive because of having cell walls of a thick peptidoglycan layer (50–90%) which stained violate under the microscope; on contrast, a thinner layer of peptidoglycan (less than 10%) was observed which stained pink was found for isolate GR and considered as gram-negative [88]. From the morphological study, LL2 and GL were found to be cocci, and LL1, LF, and GR were found to be rod-shaped bacteria.

LL2 and GL (gram-positive cocci) showed positive results for catalase, oxidase, and methyl red tests (**Table 6**) which implied that the isolates were either *Staphylococcus spp*. or *Micrococcus spp*. [66, 89]. Following negative results for the indole test and VP test (**Table 6**), LL2 showed positive results for mannitol salt fermentation (**Table 6**) whereas GL showed a negative result. Thus, LL2 is strongly assumed to be *Staphylococcus aureus* and GL as *Micrococcus luteus* [65, 90]. Gram-positive rods LL1 and LF showed positive results for catalase, oxidase, and methyl red tests, as well as negative results for indole tests, VP tests, and mannitol salt fermentation (**Table 6**). Therefore, these two isolates were considered similar to *Bacillus spp*. [66, 91]. Positive test results in the citrate utilization test (**Table 6**) conferred isolate LL1 to be similar to *Bacillus subtilis*. On the other hand, the gram-negative rod-shaped isolate GR showed

**Table 12. Physicochemical properties of the binding compounds.**

| Lipinski filters analysis | | | | | | |
|---|---|---|---|---|---|---|
| **Lipinski filters** | **Lopinavir** | **Aureusimine** | **Bacitracin** | **Carotenoid** | **Microansamycin** | **Staphyloferrin** |
| **Mol. weight (g/mol)** | 628.80 | 244.29 | 1422.69 | 568.87 | 319.35 | 480.38 |
| **Num. heavy atoms** | 46 | 18 | 100 | 42 | 23 | 33 |
| **Num. rotatable bonds** | 17 | 3 | 35 | 14 | 0 | 18 |
| **Num. H-bond acceptors** | 5 | 3 | 20 | 2 | 5 | 14 |
| **Num. H-bond donors** | 4 | 2 | 17 | 1 | 2 | 9 |
| **Molar Refractivity** | 187.92 | 70.91 | 399.90 | 188.21 | 84.24 | 101.10 |
| **Druglikeness** | | | | | | |
| **Lipinski Filter** | Yes | Yes | No | No | Yes | No |
| **Ghose** | No | Yes | No | No | Yes | No |
| **Veber** | No | Yes | No | No | Yes | No |
| **Egan** | Yes | Yes | No | No | Yes | No |
| **Muegge** | No | Yes | No | No | Yes | No |
| **ADMET analysis (Probability)** | | | | | | |
| **a) Absorption** | | | | | | |
| **Blood–Brain Barrier** | + (0.9104) | +0.9670 | +0.9270 | +0.9049 | +0.9435 | +0.9640 |
| **Human Intestinal Absorption** | + (0.9624) | +0.9925 | +0.8746 | +0.9879 | +0.9063 | -0.6694 |
| **Bioavailability Score** | 0.55 | 0.55 | 0.17 | 0.17 | 0.55 | 0.11 |
| **Caco-2** | + (0.9313) | +0.7591 | -0.8675 | -0.7841 | +0.5576 | -0.8885 |
| **P-glycoprotein Substrate** | + (0.7009) | -0.8841 | +0.8733 | -0.6950 | -0.5051 | -0.7300 |
| **P-glycoprotein Inhibitor** | + (0.9511) | -0.9197 | +0.7422 | +0.8359 | -0.9012 | -0.5469 |
| **b) Distribution** | | | | | | |
| **Subcellular localization** | Mitochondria (0.7846) | Mitochondria (0.8990) | Mitochondria (0.4232) | Mitochondria 0.7289 | Mitochondria 0.7657 | Mitochondria 0.8125 |
| **c) Metabolism** | | | | | | |
| **CYP1A2 inhibition** | -(0.8935) | +0.6112 | -0.9046 | -0.8585 | -0.8444 | -0.8870 |
| **CYP2C9 inhibition** | -(0.7326) | -0.8667 | -0.9071 | -0.8478 | -0.8855 | -0.9408 |
| **CYP2C19 inhibition** | -(0.7983) | -0.5533 | -0.9025 | -0.8443 | -0.8605 | -0.9215 |
| **CYP2D6 inhibition** | -(0.9438) | -0.9369 | -0.9231 | -0.8980 | -0.9441 | -0.9250 |
| **d) Toxicity** | | | | | | |
| **AMES Toxicity** | -(0.8300) | -0.8400 | -0.6900 | -0.8300 | -0.7800 | -0.7700 |
| **Carcinogens** | -(0.6710) | -0.9286 | -0.8714 | -0.6888 | -0.9286 | -0.9571 |
| **Acute Oral Toxicity (kg/mol)** | 2.994 | 1.301 | 3.098 | 2.967 | 3.711 | 2.264 |
| **Hepatotoxicity** | +(0.7000) | +0.6750 | -0.6250 | +0.5750 | -0.7250 | -0.6250 |
| **Aqueous solubility (LogS)** | -3.414 | -2.734 | -3.079 | -2.078 | -3.091 | -1.813 |
| **e) Pharmacokinetics** | | | | | | |
| **GI absorption** | High | High | Low | Low | High | Low |
| **Log Kp (skin permeation) cm/s** | -5.93 | -6.45 | -17.88 | -1.14 | -8.52 | -12.35 |

negative results for all biochemical tests except the oxidase test. Thus, it is assumed that the isolate was similar to *Pseudomonas spp.* or *Aeromonas spp*. [66].

Antibiotics are natural or semisynthetic drugs used against bacterial infections. A number of antibiotics are in use for medications for a long time; hence, rapid and uncontrolled uses of antibiotics lead to antibiotic resistance. New pathogens have evolved to be resistant to antibiotic action, which creates the need to discover new novel antibiotics [92, 93]. An antibiotic

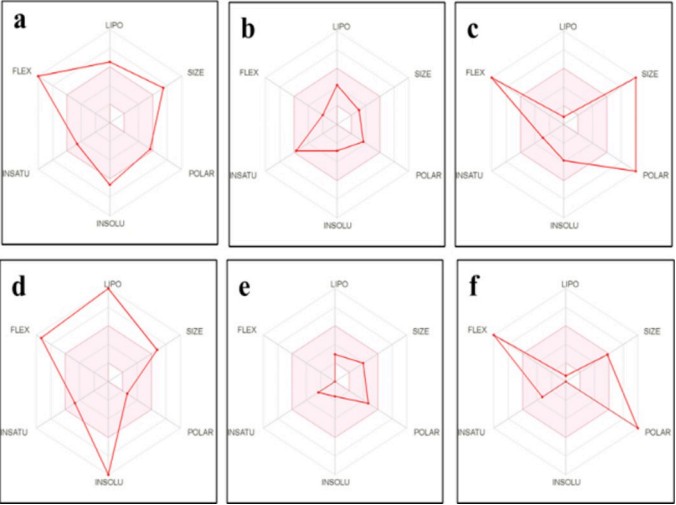

**Fig 14. Summary of the pharmacokinetic properties of the compounds.** (a) Lopinavir, (b) aureusimine, (c) bacitracin, (d) carotenoid, (e) microanamycin, (f) staphyloferrin The color space is a suitable physiochemical space for oral bioavailability. LIPO Lipophility: -0.7 < XLOGP3 < +5.0. SIZE: 150 g/mol: < MW < 500 g/mol. POLAR (Polarity): 20Å2 < TPSA < 130 Å 2. INSOLU (insolubility): 0 < Log S (ESOL) < 6. INSATU (insaturation): 0.25 < Fraction Csp3 < 1. FLEX (Flexibity): 0 < Num. rotatable bonds < 9.

sensitivity test was performed to identify suitable antibiotics that will be most effective against the specific type of bacteria. Here, we performed this test to evaluate the most commonly used antibiotics against bacterial isolates to determine the harmful bacterial strains. Ten frequently used antibics containing discs (Himedia, India), penicillin G (P), chloramphenicol (C), ampicillin (AMP), streptomycin (S), vancomycin (VA), norfloxacin (NX), tetracycline (TE), nalidixic acid (NA), erythromycin (E), and ceftriaxone (CTR), were used to identify their sensitivity against our five isolated endophytic bacteria (**Fig 16**). Each isolate showed resistance (R) toward penicillin G. Isolate LL2 also had the highest resistance percentage by showing resistance to tetracycline, erythromycin, and ceftriaxone. LL1 expressed intermediate sensitivity (I) to ampicillin. The rest of the antibiotics were found to be sensitive (S) to other isolated endophytic bacteria. One of the major causes of sensitivity to antibiotics can be the reduced exposure of endophytic bacteria in the human system rather than the inner plant [39, 94, 95].

16S rDNA amplification was conducted through PCR for further confirmatory identification of isolated bacterial species. Genomic DNA was extracted by the conventional method [96], and DNA concentrations were measured by Nanodrop spectroscopy (Thermo Scientific, USA). Then the extracted DNA of five bacterial isolates was amplified by universal 16S rDNA primer pairs [5, 68]. All five isolates showed prominent bands (~800 bp) compared to the 50 bp DNA marker on a 2% agarose gel (**Fig 2**). In general 16S rDNA and Sanger sequencing (Macrogen's sequencing service, South Korea) were performed to identify the strains of the respective bacterial isolates [97]. Homology analysis inferred from 16S rDNA sequence comparison clearly verified that the five isolates clustered with *Priestia megaterium* (LL1), *Staphylococcus caprae* (LL2), *Neobacillus drentensis* (LF), *Micrococcus yunnanensis* (GL), and *Sphingomonas paucimobiliz* (GR) (**Table 6**) having 99% identity in BLAST analysis [18]. All the results were submitted to GenBank where the accession numbers MW494406 (LL1), MW494408 (LL2), MW494401 (LF), MW494402 (GL), and MZ021340 (GR) were assigned.

Phylogenetic trees were constructed (**Fig 3**) to determine the similarity with other members of the strains found in homology analysis [71]. Evolutionary analyses were conducted in MEGA 11 [50] using the neighbor-joining method. Only sequences from the type of mostly

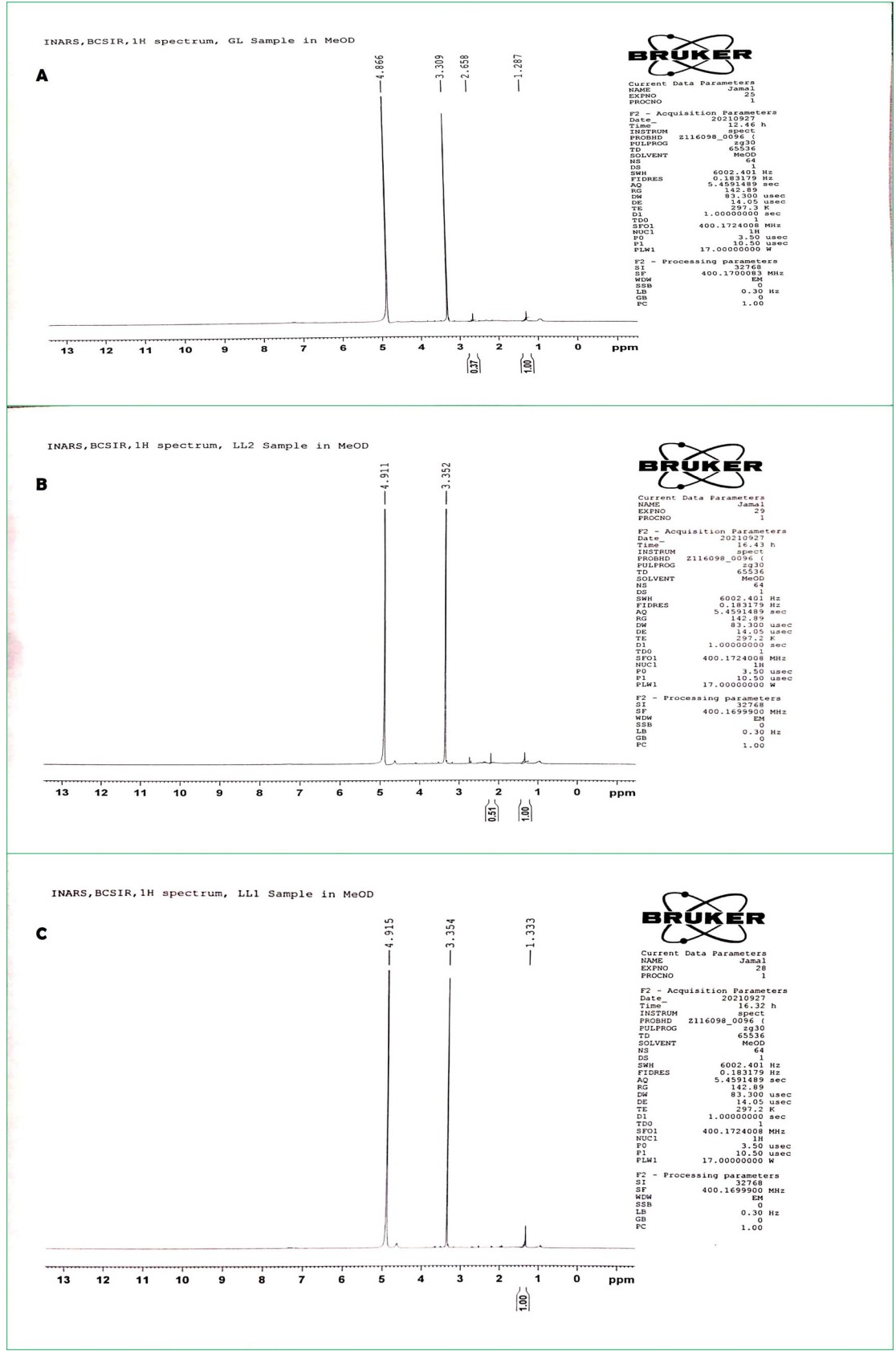

**Fig 15. NMR 1H spectroscopy result: In the left the metabolite structure and on the right 1H NMR spectrum.** Every peak is responsible for a respective different proton type. (A) Microansamycin, (B) Aureusimine, and (C) Surfactin.

identical strains were taken into account. More than 200 bacterial genera from 16 phyla have been reported to be associated with endophytes, with the majority of the species belonging to the phyla Actinobacteria, Proteobacteria, and Firmicutes [98]. Our study identified three Firmicutes (*Priestia megaterium*, *Staphylococcus caprae*, *and Neobacillus drentensis*), one actinobacterial (*Micrococcus yunnanensis*), and one proteobacterial (*Sphingomonas paucimobiliz*) endophyte from different medicinal host plants around the Chittagong University campus.

Endophytic bacteria have been reported to play a vital role in growth promotion, nutrient management, disease control, and biotic and abiotic stress tolerance in food and crop plants [9, 99]. Phytohormones play a significant role in plant growth promotion and it was suggested that IAA production helps the plants grow root parts, which increases the plants' nutrient uptake [99]. Five identified isolates were screened for growth-promoting parameters and found positive for indole acetic acid (IAA) production. Quantitative estimation of all isolates was performed in ISP2 broth medium and production ranged from 14.26 μg/mL to 25.561 μg/mL (**Table 7**). The maximum IAA yield was observed in *Sphingomonas paucimobiliz* (25.561 μg/mL). This finding is similar to Khamna et al. (2009) who stated that IAA production levels ranged from 13.73 μg/ml to 142 μg/ml [71, 100] in bacterial endophytes. Ammonia production is an indirect mechanism of plant growth promotion and can play a significant role in suppressing phytopathogens [101]. In the present study, all isolates showed ammonia production in peptone water broth ranging from 5.3 mg/mL to 24.98 mg/mL (**Table 7**). Qualitative estimation showed that isolating *Staphylococcus caprae* resulted in the maximum yield of ammonia (24.98 mg/mL). Studies have suggested similar results of ammonia production range

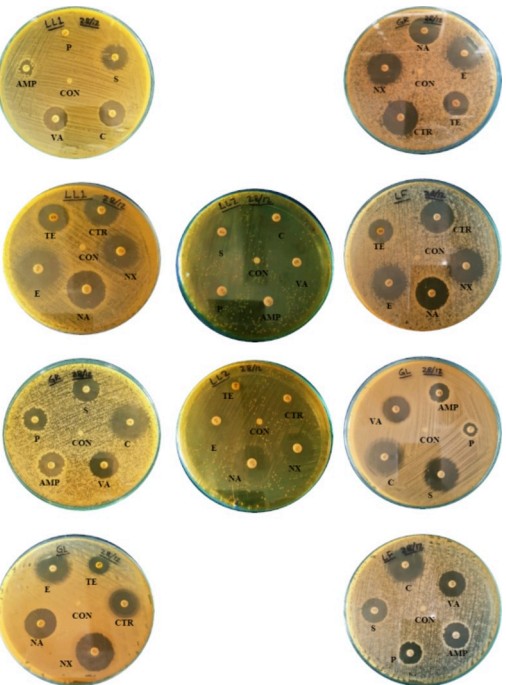

**Fig 16. Antibiotic sensitivity test of isolates showing inhibition zones (mm).** Control (CON) indicates discs without antibiotics in each case. Disc codes are used to show the antibiotic discs for every isolate.

by endophytic bacteria and inferred an association in plant growth promotion [6, 71]. The isolated bacterial strains *Priestia megaterium* (LL1) and *Staphylococcus caprae* (LL2) showed protease positive results by degrading casein in skim milk agar (**Fig 5**). Previous studies suggested that *Bacillus*, *Paenibacillus*, and *Pseudomonas* are highly recognized as protease producers [8, 63].

Phenolics and flavonoids are responsible for exhibiting the antioxidant properties of chemical compounds. The presence of antioxidant compounds such as total phenolic content (TPC) and total flavonoid content (TFC) in endophytic bacteria and solvent extracts was assessed [89] in the present study. As an extraction solvent, ethyl acetate extract (EA) was used because it is selective in extracting low molecular weight phenols and flavonoids [70]. The highest TPC value was found in *Staphylococcus caprae* (406.653±0.01 µg/mL) (**Table 7**) and it also exhibited the highest TFC value (45.18±0.5 µg/mL) (**Table 7**). The TPC and TFC values found in the present study were higher than those reported in previous studies [70, 102, 103]. Therefore, endophytic *Staphylococcus caprae* could be a source of natural phenolics and flavonoids to act as antioxidant compounds. These compounds also shows antiviral activity against various microorganisms [104, 105]. Moreover, some recent studies presented the efficacy of these compounds against SARS-COV-2 [106].

In recent years, bioactive compounds have been in high demand due to their benefits to humans and plants in various sectors of application. Thus, endophytic bacteria act as a promising resource of biotechnologically valuable bioactive compounds as well as secondary metabolites [107, 108].

Most bacterial secondary metabolites are produced by biosynthetic gene clusters consisting of key enzymes, such as polyketide synthases (PKSs) or non-ribosomal peptide synthetases (NRPSs), and contiguous genes encoding tailoring enzymes and transporters [6]. To explore the biosynthetic gene clusters and their associated bioactive secondary metabolites from our isolated bacterial strains, we used the webserver tool antiSMASH 5.0 bacterial version [109]. With default parameters of input data, we found thirteen different secondary metabolite regions from five isolates (**Fig 6**). The most abundant regions found were Terpene, Siderophore, and Type III polyketide synthase regions followed by NRPS, Lanthipeptide, and Betalactone regions (**Fig 7**). Among the five isolates, the *Priestia megaterium* strain showed the maximum number of metabolite regions, conferring itself as a potent source of bioactive compounds [19]. The antiSMASH server linked the data with the MIBig server, where we found the information of the most similar known cluster related to the database and represented the features of predicted bioactive compounds [110]. Nine compounds were found to be matched with the database, and structure, features and the list of core biosynthetic genes along with additional and other regulatory genes were retrieved from the results (**Table 9**). Four nonribosomal peptide (NRP)-type compounds–Surfactin (lipopeptide), Bacitracin, Microansamycin (cyclic despeptide), and Aureusimine–Three terpene type compounds–Carotenoid, Stenothricin, and Zeaxanthin–were identified. The other two compounds were staphyloferrin **A** (siderophore) and fengycin (betalactone).

Compound results are represented as the most similar known compounds from the BLAST result of the database, in which aureusimine, staphyloferrin A, and zeaxanthin were matched 100% with our query sequences (**Fig 8**). Thus, *S. caprae* and *S. paucimobiliz* can be a great source of antioxidant metabolite extraction for further detailed study of their bioactivity.

Compound structure, features, and information on core and biosynthetic genes involved in the secondary metabolite clusters were also retrieved from the MIBiG server [26]. Later, the genes and protein products were evaluated through the UniProt server to obtain information on proteins, their biological processes, and their biosynthesis pathways (**Table 10**). Amino acid sequences from the clusters were run through the peptide search of UniProt and the

protein products of the genes were extracted. After that, the STRING server (version 11.0) [72] was used to represent the networks, association, and correlation between the genes and proteins involved in biosynthetic gene clusters [17, 111] (**Fig 9**). Unfortunately, enough protein data could not be retrieved for aureusimine, microansamycin, and stenothricin from the database to show the interactions.

**Surfactin** is widely known as a biosurfactant along with its antimicrobial properties and has already shown a handful of bioactive potentialities [74]. Defining the genes involved in the surfactin cluster, we found that *srfAA*, *srfAB*, *srfAC*, and *srfAD* were the core genes involved in surfactin biosynthesis (**Table 10**). *srfAD* is the core gene associated with synthetase genes and the transporter gene *tcyC* to modulate the surfactin biosynthesis pathway. It was shown to be a clear protein–protein interaction of these genes. On the other hand, the additional biosynthetic genes *ycxB* and *ycxC*ad*ycxD* encode regulation- and transport-related proteins involved in the processing of this secondary metabolite. Surfactin has shown experimental success due to its antibacterial capacity [112] and inhibitory mechanism against pathogenic bacteria [113]. It has been shown to enhance surfactin production from *B. licheniformis* and *Bacillus amyloliquefaciens* [114], and thus, it is promising to use *Priestia megaterium* as a potential source to synthesize this biosurfactant.

**Bacitracin** is a cyclic polypeptide antimicrobial derived from the bacterium *Bacillus subtilis* that has wide use for the treatment of superficial bacterial skin infections [75]. The bacitracin gene cluster comprises three core biosynthetic genes (*bacA*, *bacB*, and *bacC*) and three additional genes (*bacR*, *bacS*, and *bacT*) (**Table 10**). From the protein database, it was found that the core bac genes are directly involved in the antibiotic (bacilysin) biosynthesis pathway, conferring that *P. megaterium* possesses such a synthesis pathway. This isolate can be used for metabolic engineering to enhance bacitracin action, as it is now graded as less functional against multidrug-resistant bacteria. No data involved in using endophytic bacteria using bacitracin production have been found previously, and thus, the metabolic engineering strategy to enhance bacitracin production is attractive.

**Carotenoids**, a subfamily of isoprenoids, are among the most widespread, diverse classes of all-natural products and biomolecules. Bacteria synthesize isoprenoids from isopentenyldiphosphate (IPP) and its isomer dimethylallyldiphosphate (DMAPP) through the methylerythritol 4-phosphate (MEP) pathway [115]. The *crtM* gene is the core gene that initiates the biosynthesis pathway [116], which clarifies the central action of the protein network (**Table 10**). Carotenoids are well-known pigments in plants, algae and photosynthetic bacteria. It is also an antioxidant for humans consumed by foods and fruits.

**Zeaxanthin** is also a carotenoid essential for our eyes against photooxidative damage from UV light. Animals and humans cannot synthesize zeaxanthin, and thus, it must be obtained through diet [117]. Zeaxanthin is supplemented through food, and now, it has increased the commercial demand for production [118]. Endophytic bacteria can be a reliable source for commercially synthesized carotenoids such as zeaxanthin and from this research, the endophytic isolate *S. paucimobiliz* can be used as a novel resource for metabolite production [119].

**Fengycin** is a novel antimicrobial agent effective against the fungal community isolated from *B. subtillis* [93]. Twelve different genes distributed in 4 synthetic clusters are responsible for the synthesis of this metabolite. A glycoside hydrolase (*yngK*) plays a connector role in the regulation of neighborhood nonribosomal synthetase activity (*fenA*, *fenB*, *fenC*, *fend*) and regulation activity (*yngE*, *yngF*, *yngG*, *yngH*, *yngI*, *yngJ*) of the genes involved in the fengycin synthase pathway (**Table 10**). Fengycin exhibited strong antifungal activity and inhibited the growth of several plant pathogens, particularly many filamentous fungi [120]; thus, the isolate *Neobacillus drentennsis* can be utilized as a harbor of novel antifungal secondary metabolites. Last, the siderophore compound staphyloferrin A comprises 4 core biosynthetic genes related

to nonribosomal peptide synthetase in the action of siderophore biosynthesis, an important mechanism to overcome iron limitation [121]. It has been used as a detector of multidrug-resistant *Staphylococcus aureus* (MRSA) [6] and can open a new window of research on endophytic bacteria against antibiotic resistance mechanisms.

Most biologically active polyketide and peptide compounds are synthesized by polyketide synthases (PKSs) and nonribosomal peptide synthetases (NRPSs), which have been widely utilized to assess the biosynthetic capacity of culturable and nonculturable bacteria [19, 114, 122]. PKS and NRPS domain occurrence was analyzed for our isolated bacterial strains from the SBSPKS V2 database. The identified strains showed significant PKS and NRPS domain data for evaluating endophytes of biosynthetic potential. Therefore, *Bacillus spp*. represented the highest percentage (PKS 36%, NRPS 72%), whereas *Staphylococcus spp*. (PKS 22%, NRPS 24%), *Micrococcus spp*. (PKS 13%, NRPS 17%) and *Sphingomonas* spp. (PKS 21%, NRPS 11%) having a moderate percentage (**Fig 10**). This result demonstrated the universal distribution of these domains in terms of the identification of biosynthetic genes for secondary metabolite production [15, 56, 123].

The presence of biosynthetic genes encoding polyketide synthases (PKS), nonribosomal peptide synthetases (NRPS), 1-aminocyclopropane-1-carboxylate deaminase (ACCD), and chalcone synthase (CHS) was analyzed by identifying five strains using four sets of degenerate primers (**Table 5**). PCR amplification of all five isolates showed a band of the expected size (~700–800 bp) for the NRPS gene (**Fig 11B**). PKS candidate amplicons (~700–800 bp) were detected in *Priestia megaterium* (LL1), *Staphylococcus caprae* (LL2), *Micrococcus yunnanensis* (GL), and *Sphingomonas paucimobiliz* (GR) (**Fig 11A**), whereas *Sphingomona paucimobiliz* (GR) showed only positive amplification of the ACCD gene (**Fig 11C**). For the CHS gene, no amplicons were found for any endophytic strain (**Fig 11D**). A putative prospect for bioactive secondary metabolites indicates that the positive strains can be a good candidate to research for natural bioactive compounds [12, 124]. *Sphingomonas paucimobiliz* showed positive results for three of the genes except for ACCD, which indicates its involvement in the pathways of plant-endophytic interaction and may be a reservoir of secondary metabolites [4, 125]. Further research and investigation of these gene and gene clusters may explore the novel bioactive natural products defining endophytic bacteria and their host interaction in medicinal plants [107, 126].

Microbial metabolites have some major advantages over synthetic drugs. Following the failure of numerous conventional medicines to treat viral infections and the emergence of particular viral resistances, interest in microbial metabolites as potential antiviral agents has grown. The recent COVID pandemic also had an immense impact on sorting out the best possible compounds against the deadly coronavirus. To exploit the probable candidate against SARS-COV2, predicted bioactive secondary metabolites were subjected to molecular docking against 3-chymotrypsin-like protease (3CL$^{pro}$) of SARS-COV2. Out of nine metabolites, five metabolites were used to dock against the target protein, and four of them (surfactin, stenothricin, fengycin, and zeaxanthin) were excluded due to the incomplete conformation of their structure. The results from this study revealed that lopinavir, the reference inhibitor, had a binding affinity of −7.2 Kcal/mol for 3CL$^{pro}$ of SARS-COV-2 (**Table 11**). The two top docked compounds to SARS-COV-2 3CL$^{pro}$ are Microansamycin (-7.7 kcal mol$^{-1}$) and Aureusimine (−6.0 kcal. mol$^{-1}$). The results obtained from the ligand-protein binding interaction showed that lopinavir docked into the receptor-binding site and interacted via a conventional hydrogen bond and Pi-alkyl bond to LYS137. It further interacted with GLU290 via a Pi-Anion bond. This was attributed to multiple noncovalent interactions, such as hydrogen bonds and van der Waals (VDW) interactions, with other amino acid residues (GLU288, THR169, GLY170, VAL171, HIS172, SER139, GLY138, TYR126, GLN127, CYS128, LYS5) at the active site of 3CLpro (**Fig 13A**).

Microansamycin, the topmost docked compound to 3CL^pro of SARS-COV-2, interacted via three conventional hydrogen bonds to the GLN107, THR111, and SER158 residues, along with VDW noncovalent interactions with ILE106, GLN110, PHE294, ILE152, PHE, ASN151, ASP153, and VAL104 amino acid residues (**Fig 13B**).

Aureusimine interacted with PHE294 via a Pi-Pi stacked interaction along with four conventional hydrogen bonds, THR111, ASP153, LYS102, and SER158, of 3CL^pro. The other VDW bonds were ASP295, THR292, GLN110, ASN151, and VAL104 (**Fig 13C**).

These with a possible inhibitory propensity against the SARS coronavirus were identified based on the findings of these docking results. The stronger interactions of these two with 3CLpro compared to the reference compound imply that they may alter the viral protease function, which is required for the processing of viral replicase polyproteins.

To validate our prediction of being a potential drug, we analyzed the pharmacokinetic properties of the docked compounds (**Table 12**). According to Lipinski's rule, an orally active medication cannot contain more than one violation of the following conditions: Not >5 hydrogen bond donors (oxygen or nitrogen atoms with one or more hydrogen atoms); Not >10 hydrogen bond acceptors (nitrogen or oxygen atoms); A molecular mass <500 Daltons; and an octanol-water partition coefficient (logP) not greater than 5. Microansamycin and aureusimine fulfilled the requirements with corresponding favorable predicted ADMET parameters (**Fig 14**). The five compounds' anticipated filtering analyses revealed characteristics that point to favorable ADME/tox and pharmacokinetic capabilities. This also suggests that the best-docked compounds (microansamycin and aureusimine) have drug potentiality. However, we have found microansamycin, aureusimine, and stenothricin through NMR results (**Fig 15**) that were present in the bacterial crude extract. Both microansamycin and aureusimine show various bioactivities including antibacterial, antiviral, antifungal, antitumor, and immunosuppressive [127, 128]. Hence, our study also revealed the antiviral potentiality of those metabolites by using different computational tools. This might be a harbinger of a structure-based medication design for COVID-19 that targets the 3CL^pro of SARS-COV-2. Therefore, antiviral microbial metabolites may present a significant opportunity in the field of pharmaceutical research and development in the near future, and endophytic bacterial bioactive compounds can be a rich source of natural products.

## Conclusion

This study revealed that endophytic bacterial secondary metabolites could be used as valuable bioactive compounds against disease-causing pathogens. Our study also suggests that the endophytic bacterial bioactive compounds microansamycin and aureusimine expose pharmaceutical efficacy targeting 3CLpro of SARS-COV-2 which represents a potential ingredient for pharmaceutical applications with antiviral properties and are worthy of future study for medication against SARS-COV-2.

## Supporting information

**S1 Raw images. Electrophoretic separation (2% agarose) of the 16S rDNA gene of different isolates.**
(TIF)

**S2 Raw images. PCR amplifications of biosynthetic genes.** (C) ACCD gene.
(TIF)

**S3 Raw images. PCR amplifications of biosynthetic genes.** (D) CHS gene (No Result).
(TIF)

**S4 Raw images. PCR amplifications of biosynthetic genes.** (B) NRPS gene.
(TIF)

**S5 Raw images. PCR amplifications of biosynthetic genes.** (A) PKS gene.
(TIF)

**S1 Fig. All the compounds' structures in 3D.**
(DOCX)

## Acknowledgments

This study was partially supported by the Research & Publication Cell of the University of Chittagong, Bangladesh, and the Ministry of Science and Technology, Government of the peoples' republic of Bangladesh (Special Allocation). This research was done in the 'Molecular Biology Laboratory' of the Department of Genetic Engineering and Biotechnology, University of Chittagong. The authors are pleased to mention the valuable suggestions during this research from Professor Kazuyuki Shimizu, Department of Bioscience & Bioinformatics, Kyushu Institute of Technology, Iizuka, Fukuoka 820–8502, Japan, and Institute of Advanced Bioscience, Keio University, Tsuruoka, Yamagata 997–0017, Japan. The research team is pleased to acknowledge the Bangladesh Council of Scientific and Industrial Research (BCSIR), DHAKA, Bangladesh for NMR detection.

## Author Contributions

**Conceptualization:** Lolo Wal Marzan.

**Data curation:** Yasmin Akter, Rocktim Barua, Md. Nasir Uddin.

**Formal analysis:** Abul Fazal Muhammad Sanaullah, Lolo Wal Marzan.

**Funding acquisition:** Lolo Wal Marzan.

**Investigation:** Yasmin Akter, Rocktim Barua, Md. Nasir Uddin.

**Methodology:** Yasmin Akter, Rocktim Barua, Md. Nasir Uddin.

**Supervision:** Yasmin Akter, Lolo Wal Marzan.

**Writing – original draft:** Rocktim Barua.

**Writing – review & editing:** Yasmin Akter, Md. Nasir Uddin, Lolo Wal Marzan.

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
