## [Decision Letter · Decision Letter 0]

11 Feb 2022

PONE-D-22-01192In silico potentiality of secondary metabolites from endophytic bacteria against SARS-COV-2.PLOS ONE

Dear Dr. Marzan,

Thank you for submitting your manuscript to PLOS ONE. After careful consideration, we feel that it has merit but does not fully meet PLOS ONE’s publication criteria as it currently stands. Therefore, we invite you to submit a revised version of the manuscript that addresses the points raised during the review process.

We look forward to receiving your revised manuscript.

Kind regards,

Milkyas Endale, PhD

Academic Editor

PLOS ONE

Journal Requirements:

Reviewers' comments:

Reviewer's Responses to Questions

**Comments to the Author**

1. Is the manuscript technically sound, and do the data support the conclusions?

Reviewer #1: Partly

Reviewer #2: Partly

2. Has the statistical analysis been performed appropriately and rigorously? 

Reviewer #1: Yes

Reviewer #2: N/A

3. Have the authors made all data underlying the findings in their manuscript fully available?

Reviewer #1: Yes

Reviewer #2: Yes

4. Is the manuscript presented in an intelligible fashion and written in standard English?

Reviewer #1: No

Reviewer #2: No

5. Review Comments to the Author

Reviewer #1: Comments on Manuscript Number PONE-D-22-01192

In silico potentiality of secondary metabolites from endophytic bacteria against SARS-COV-2

Major Concerns-

1. Although authors of manuscript are claiming that their study is related with potentiality of secondary metabolites from endophytic bacteria against SARS-COV-2, but this claim seems far from reality after observing following facts-

(a) Authors have been started collecting plant samples since 15/07/2019 and 9 sample out of 16 were collected before Dec. 2019 which they referred as the date of first case of covid-19. So, it is quite unreliable that the study was started with the aim to produce potential candidates against SARS-COV-2.

(b) Surprisingly, after doing rigorous experimentation on 16 plants and 48 plant parts they got positive results only for five plants which they started since Feb. 2020 and so on. It is giving an indication that 9 plants which have been reported in manuscript before Feb. 2020 are just to enhance data in the manuscript, although this indication can be completely false but impression of the results is like this only.

(c) Authors have been claimed activity against SARS-COV-2, the ligand they used for docking is based on their prediction of secondary metabolites from different servers (antiSMASH etc…). But the major concern in this study is that, they have been characterized the bacterial species by experimentation but after that everything is just based on prediction.

(d) On first stage they predicted the secondary metabolites present in their characterized bacterial species. Predictions may be far from reality.

(e) But still if we assume that these predicted secondary metabolites are very near to the reality (which is generally not true), further they predicted their activity against SARS-COV-2 through molecular docking and drug likeness properties. The major concern in this docking protocol is that the research and available database in the servers is far too limited to claim promising results as most of the researches are in primitive level through out the world. If one wants to predict for the disease which is existed for the last many years then due to sound database available online; these database servers and software’s mostly show promising results which is rarely possible in the case of SARS-COV-2. So, if authors want to claim it, they should first find out true secondary metabolites rather than predicted and may enhance the work further. Otherwise, they may change claim of their work via rewriting the manuscript accordingly.

(f) Lines 138-143- “It was initially detected in December 2019 in Wuhan, China [31], and on 11 March 2020, the outbreak was declared by the World Health Organization (WHO) as a global pandemic that has caused a total death of 4,644,740 worldwide as of September 15, 2021, with nearly 4 million new cases reported globally in the past week (6-12 September) according to the World Health Organization. Unfortunately, there is no particular medication or therapy for COVID-19 at this time.” We received your manuscript in Jan. 2022 and these lines are extremely confusing and again indicating that the study may not aimed towards SARS-COV-2.

2. Authors are informing that “Five biochemically identified bacterial isolates were then sent for sequencing (Macrogen, South Korea).” This is the most important part of their study which they outsourced from South Korea. This is a major concern of the study.

3. As per table 6 they provisionally identified bacterial strains but actual findings are far from their assumptions, so how they will counter this gap in the study?

4. Introduction and discussion part is far too long concerning research article and must be rewritten thoroughly and concisely.

5. Authors are claiming “Determination of antioxidant compounds from crude extracts”, but in the discussion part they did not mention the importance and future prospects of this study.

6. Figure 15 is not self-explanatory, they referred it in Drug likeness and surely it is but it is not understandable without any explanation. Authors may omit this figure or should explain it thoroughly.

Minor concerns-

1. Lines 68-71 “Natural compounds from endophytes exhibit activity as antimicrobials, antifungals, anticarcinogens, immunosuppressants, or antioxidants; thus, secondary metabolites produced by endophytes can be implemented as therapeutics in the pharmaceutical and agricultural industries [3,4]”

Separate references are needed to claim every other activity (primary reference must be used). The same comment applied for lines 100-102, 119-120, 126-127, 541-543.

2. In Table-2 Location (around CU campus) is given as separate entry which I feel is not much important for international research community, it may be useful for local people available there in author’s country.

3. At many places English grammar is not correct and should be checked again thoroughly.

4. Line 395- heading must be in uniform font and style.

5. Lines 537-539 “Therefore, one of the major objectives of this study is to isolate and characterize endophytic bacteria from our local medicinal plants to study their secondary metabolites, which have a wide range of applications in our health sector”, rather than only concentrate on their health sector, Authors should show their concern globally as they are claiming study related with a global pandemic.

Reviewer #2: The authors presented interesting research area. The in silico techniques the authors employed are to be appreciated. However, the manuscript needs more modification to meet PLOS ONE’s requirements.

1. It lacks consistency of ideas.

2. Scientific ideas are appeared without citation or the authors used a reference for three to six lines of scientific ideas. It should be re written

3. The ideas between line numbers 89 and 90, that of 91 has no relationship. It is not clear how the authors write the first two lines.

4. Surprisingly the authors used wrong references. Example the reference used to cite SwissADME in the method section of the manuscript is totally wrong.

5. The method section of the presented manuscript suffers from appropriate method writing and citations.

6. The authors reported that Microansamycin (-7.7 kcal mol-1) and Aureusimine (-6.0 kcal.mol-1) are good against the studied target. However, they are not found to show interaction with Cys amino acid residue. How could they suggest the compounds are good against the studied target without interaction with Cys amino acid?

7. Abstract and conclusions are not presented in an appropriate fashion with data support.

I suggest this paper has to be resubmitted.

6. PLOS authors have the option to publish the peer review history of their article (what does this mean?). If published, this will include your full peer review and any attached files.

Reviewer #1: **Yes: **Dr. Ankita Garg

Reviewer #2: No

---

## [Author Response · Author response to Decision Letter 0]

2 Apr 2022

Response to reviewer #1

We thank the honorable reviewer #1 for valuable feedback. Please find our response below-

Comments on Manuscript Number PONE-D-22-01192: ‘In silico potentiality of secondary metabolites from endophytic bacteria against SARS-COV-2’

Major Concerns-

1. Although authors of manuscript are claiming that their study is related with potentiality of secondary metabolites from endophytic bacteria against SARS-COV-2, but this claim seems far from reality after observing following facts- 

(a) Authors have been started collecting plant samples since 15/07/2019 and 9 sample out of 16 were collected before Dec. 2019 which they referred as the date of first case of covid-19. So, it is quite unreliable that the study was started with the aim to produce potential candidates against SARS-COV-2. 

Response: The authors would like to thank the reviewer for pointing out the issues and helping us to improve its quality.

Our initial objective was to characterize and identify endophytic bacteria which can secrete secondary metabolites including medicinal value against microbial diseases. Subsequently, we found the significant medicinal value of those metabolites after diverse analysis. During our study Covid-19 pandemic starts, thus our interest and focus have been amended to find if those metabolites had any medicinal value against SARS-COV-2 including other pathogenic viruses.

(b) Surprisingly, after doing rigorous experimentation on 16 plants and 48 plant parts they got positive results only for five plants which they started since Feb. 2020 and so on. It is giving an indication that 9 plants which have been reported in manuscript before Feb. 2020 are just to enhance data in the manuscript, although this indication can be completely false but impression of the results is like this only. 

Response: We would like to thank the reviewer for these valuable comments. 48 plant parts have been selected to identify and isolate endophytic bacteria during experiments. Then, the plant extracts derived from 48 plant parts were spread over the solid Nutrient Agar (NA) media. Subsequently, several bacterial colonies were found and finally, sixteen pure colonies were selected (Table 3) as well as isolated after doing rigorous experiments. So, all (48) collected plant parts were used throughout the experiment.

Consequently, we have collected samples during different seasons (15/07/2019 and Feb. 2020) to compare the prevalence of antimicrobial effects. Besides, an epidemiological study among large samples might give significant results, that’s why we increased our samples.

(c) Authors have been claimed activity against SARS-COV-2, the ligand they used for docking is based on their prediction of secondary metabolites from different servers (antiSMASH etc.). But the major concern in this study is that, they have been characterized the bacterial species by experimentation but after that everything is just based on prediction. 

Response: Thanks for your valuable comments. After characterization of specific bacteria, we have identified different metabolites such as Total phenolic content (TPC), Total flavonoid content (TFC) by the colorimetric method mentioned in the result section (Table 7). To confirm specific metabolite detection we have studied different in silico metabolite predicting tools by which we predicted nine compounds. Then all bacterial crude extract was studied for NMR analysis for validation of those metabolites. By proton NMR spectroscopy, we found three metabolites (Microansamycin, Aureusimine, and, Surfactin) out of the nine predicted metabolites. The NMR data has been included in the manuscript (Fig 16). However, we could not find the other six metabolites, it may be because of the low expression of the relevant genes that are responsible for those metabolite productions.

(d) On first stage they predicted the secondary metabolites present in their characterized bacterial species. Predictions may be far from reality. 

Response: Thank you for your valuable suggestion. In the first stage, our results predicted the presence of secondary metabolites in their characterized bacterial species. According to the reviewer’s suggestions, we did NMR analysis (Fig 16), where we found our targeted metabolites (Microansamycin, Aureusimine, and, Surfactin.) which are perfectly complemented with our predicted data. 

(e) But still if we assume that these predicted secondary metabolites are very near to the reality (which is generally not true), further they predicted their activity against SARS-COV-2 through molecular docking and drug likeness properties. The major concern in this docking protocol is that the research and available database in the servers is far too limited to claim promising results as most of the researches are in primitive level throughout the world. If one wants to predict for the disease which is existed for the last many years then due to sound database available online; these database servers and software’s mostly show promising results which is rarely possible in the case of SARS-COV-2. So, if authors want to claim it, they should first find out true secondary metabolites rather than predicted and may enhance the work further. Otherwise, they may change claim of their work via rewriting the manuscript accordingly. 

Response: Thanks for your valuable suggestion. According to the reviewer’s suggestion, we have enhanced our work further to discover true secondary metabolites determined by NMR spectroscopy. And we found secondary metabolites from the NMR data which matched with our predicted secondary metabolites, and then we have already included them in our manuscript accordingly (Fig 16).

(f) Lines 138-143- “It was initially detected in December 2019 in Wuhan, China [31], and on 11 March 2020, the outbreak was declared by the World Health Organization (WHO) as a global pandemic that has caused a total death of 4,644,740 worldwide as of September 15, 2021, with nearly 4 million new cases reported globally in the past week (6-12 September) according to the World Health Organization. Unfortunately, there is no particular medication or therapy for COVID-19 at this time.” We received your manuscript in Jan. 2022 and these lines are extremely confusing and again indicating that the study may not aimed towards SARS-COV-2. 

Response: We would like to thank the reviewer for raising the issue. During our experiment, by literature review, we came to know about the antiviral potentiality of different bacterial metabolites (Table 1). When the COVID-19 pandemic situation started throughout the world, at that time, we tried to identify whether our extracted metabolites have bio-potential activity against the virus SARS-COV-2. Still, we know that there is no particular medication or therapy for COVID-19 at this time, for that purpose we tried to carry out some prediction-based as well as true experiments on the targeted drug.

However, we have deleted the confusing line (142-143)-‘Unfortunately, there is no particular medication or therapy for COVID-19 at this time.’ 

2. Authors are informing that “Five biochemically identified bacterial isolates were then sent for sequencing (Macrogen, South Korea).” This is the most important part of their study which they outsourced from South Korea. This is a major concern of the study. 

Response: Thanks for your valuable comments. All the biochemical, molecular experiments were done in our own lab. We carried out PCR by16s rDNA primer, and then the PCR product was purified in our laboratory. After that, we just sent our sample (PCR product) for DNA sequencing. Due to the unavailability of a DNA sequencing machine, we have to carry out the sequencing in South Korea. 

3. As per table 6 they provisionally identified bacterial strains but actual findings are far from their assumptions, so how they will counter this gap in the study? 

Response: Thanks for your suggestion. We know that sometimes the biochemical test cannot show 100% accurate results as a confirmatory test, therefore for further validation we sent our samples for DNA sequencing which is more reliable for the identification of bacterial species. Thus we claim the DNA sequencing result in our study is reliable. 

4. Introduction and discussion part is far too long concerning research article and must be rewritten thoroughly and concisely. 

Response: We would like to thank the reviewer for raising this issue to improve its quality. Introduction and discussion parts are rewritten thoroughly and concisely.

5. Authors are claiming “Determination of antioxidant compounds from crude extracts”, but in the discussion part, they did not mention the importance and future prospects of this study. 

Response: Thanks for your comments. In the discussion part, we included the importance and future prospects of this study.

6. Figure 15 is not self-explanatory, they referred it in Drug likeness and surely it is but it is not understandable without any explanation. Authors may omit this figure or should explain it thoroughly. 

Response: We have modified it as a self-explanatory figure.

Minor concerns- 

1. Lines 68-71 “Natural compounds from endophytes exhibit activity as antimicrobials, antifungals, anticarcinogens, immunosuppressants, or antioxidants; thus, secondary metabolites produced by endophytes can be implemented as therapeutics in the pharmaceutical and agricultural industries [3,4]” 

Separate references are needed to claim every other activity (primary reference must be used). The same comment applied for lines 100-102, 119-120, 126-127, 541-543.

Response: Thanks for your suggestion. We have modified it.

2. In Table-2 Location (around CU campus) is given as separate entry which I feel is not much important for international research community, it may be useful for local people available there in author’s country.

Response: The research work will not only be useful for local people available in our country but also will open a significant chance for carrying out future research by using this type of methodology in any international research community. Although our targeted location for the sample (medicinal plants) was around the CU (University of Chittagong) campus, the isolates collected from those medicinal plants can be found throughout the world.

3. At many places English grammar is not correct and should be checked again thoroughly. 

Response: Thanks for your suggestions. Checked and modified it accordingly.

4. Line 395- heading must be in uniform font and style. 

Response: Modified it accordingly.

5. Lines 537-539 “Therefore, one of the major objectives of this study is to isolate and characterize endophytic bacteria from our local medicinal plants to study their secondary metabolites, which have a wide range of applications in our health sector”, rather than only concentrate on their health sector, Authors should show their concern globally as they are claiming study related with a global pandemic. 

Response: Modified it accordingly.

……….

Response to reviewer #2

We would like to thank reviewer#2 for raising the following issues. 

Please find our response below on manuscript number PONE-D-22-01192: ‘In silico potentiality of secondary metabolites from endophytic bacteria against SARS-COV-2’

The authors presented interesting research area. The in silico techniques the authors employed are to be appreciated. However, the manuscript needs more modification to meet PLOS ONE’s requirements.

1. It lacks consistency of ideas.

Response: Thanks for your suggestions. We have improved and modified our manuscript as per the reviewer’s instructions.

2. Scientific ideas are appeared without citation or the authors used a reference for three to six lines of scientific ideas. It should be re written.

Response: We would like to thank the reviewer for this suggestion. We have rechecked the manuscript and corrected the error accordingly.

3. The ideas between line numbers 89 and 90, that of 91 has no relationship. It is not clear how the authors write the first two lines.

Response: We thank the honorable reviewer for valuable feedback. We have rechecked the manuscript and deleted these two lines accordingly.

4. Surprisingly the authors used wrong references. Example the reference used to cite SwissADME in the method section of the manuscript is totally wrong.

Response: We would like to thank the reviewer for this suggestion. We have modified our manuscript as per the reviewer’s instruction. Sorry for the inconvenience.

5. The method section of the presented manuscript suffers from appropriate method writing and citations.

Response: We have rechecked the manuscript and corrected the error accordingly.

6. The authors reported that Microansamycin (-7.7 kcal mol-1) and Aureusimine (-6.0 kcal.mol-1) are good against the studied target. However, they are not found to show interaction with Cys amino acid residue. How could they suggest the compounds are good against the studied target without interaction with Cys amino acid?

Response: Although Cysteine residue is not found in our targeted docking sites, other residues like asparagine, lysine, threonine, serine, etc. all are having multiple H-bond with the metabolites. That’s why it gives a strong interaction.

7. Abstract and conclusions are not presented in an appropriate fashion with data support.

Response: We would like to thank the reviewer for this suggestion. We have rechecked the manuscript and modified it accordingly.

I suggest this paper has to be resubmitted.

Response: Thanks for your suggestion. After all modifications, we have resubmitted our manuscript again.

……

---

## [Decision Letter · Decision Letter 1]

14 Apr 2022

PONE-D-22-01192R1Bioactive potentiality of secondary metabolites from endophytic bacteria against SARS-COV-2: an in-silico approachPLOS ONE

Dear Dr. Marzan,

Thank you for submitting your manuscript to PLOS ONE. After careful consideration, we feel that it has merit but does not fully meet PLOS ONE’s publication criteria as it currently stands. Therefore, we invite you to submit a revised version of the manuscript that addresses the points raised during the review process.

We look forward to receiving your revised manuscript.

Kind regards,

Milkyas Endale, PhD

Academic Editor

PLOS ONE

Journal Requirements:

Reviewers' comments:

Reviewer's Responses to Questions

**Comments to the Author**

1. If the authors have adequately addressed your comments raised in a previous round of review and you feel that this manuscript is now acceptable for publication, you may indicate that here to bypass the “Comments to the Author” section, enter your conflict of interest statement in the “Confidential to Editor” section, and submit your "Accept" recommendation.

Reviewer #1: All comments have been addressed

2. Is the manuscript technically sound, and do the data support the conclusions?

Reviewer #1: Yes

3. Has the statistical analysis been performed appropriately and rigorously? 

Reviewer #1: Yes

4. Have the authors made all data underlying the findings in their manuscript fully available?

Reviewer #1: Yes

5. Is the manuscript presented in an intelligible fashion and written in standard English?

Reviewer #1: Yes

6. Review Comments to the Author

Reviewer #1: (No Response)

7. PLOS authors have the option to publish the peer review history of their article (what does this mean?). If published, this will include your full peer review and any attached files.

Reviewer #1: **Yes: **Dr. Ankita Garg

---

## [Author Response · Author response to Decision Letter 1]

18 May 2022

Response to reviewers

We thank the honorable reviewer #1 for valuable feedback. Please find our response below-

Comments on Manuscript Number PONE-D-22-01192R1: ‘Bioactive potentiality of secondary metabolites from endophytic bacteria against SARS-COV-2: an in-silico approach’

Comments to the Author

1. If the authors have adequately addressed your comments raised in a previous round of review and you feel that this manuscript is now acceptable for publication, you may indicate that here to bypass the “Comments to the Author” section, enter your conflict of interest statement in the “Confidential to Editor” section, and submit your "Accept" recommendation.

Reviewer #1: All comments have been addressed

Response: The authors would like to thank the reviewer for pointing out all the issues in the previous round of review and helping us to improve its quality.

2. Is the manuscript technically sound, and do the data support the conclusions?

Reviewer #1: Yes

Response: We would like to thank the reviewer for this positive comment.

3. Has the statistical analysis been performed appropriately and rigorously?

Reviewer #1: Yes

Response: We would like to thank the reviewer for this positive comment.

4. Have the authors made all data underlying the findings in their manuscript fully available?

Reviewer #1: Yes

Response: We would like to thank the reviewer for this positive comment to improve its quality.

5. Is the manuscript presented in an intelligible fashion and written in standard English?

Reviewer #1: Yes

Response: We would like to thank the reviewer for this positive comment to improve its quality.

6. Review Comments to the Author

Reviewer #1: (No Response)

7. PLOS authors have the option to publish the peer review history of their article (what does this mean?). If published, this will include your full peer review and any attached files.

Do you want your identity to be public for this peer review? For information about this choice, including consent withdrawal, please see our Privacy Policy.

Reviewer #1: Yes: Dr. Ankita Garg

……….

Journal Requirements:

Response: Thanks for your suggestions. Checked and modified it accordingly.

Response: Thanks for your suggestions. Checked and modified it accordingly.

---

## [Editor Report · Decision Letter 2]

1 Jun 2022

Bioactive potentiality of secondary metabolites from endophytic bacteria against SARS-COV-2: an in-silico approach

PONE-D-22-01192R2

Dear Dr. Lolo Wal Marzan,

We are pleased to inform you that your manuscript has been judged scientifically suitable for publication and will be formally accepted for publication once it meets all outstanding technical requirements.

Kind regards,

Milkyas Endale, PhD

Academic Editor

PLOS ONE
---

## [Editor Report · Acceptance letter]

13 Jul 2022

PONE-D-22-01192R2 

Bioactive potentiality of secondary metabolites from endophytic bacteria against SARS-COV-2: an *in-silico* approach 

Dear Dr. Marzan:

I'm pleased to inform you that your manuscript has been deemed suitable for publication in PLOS ONE. Congratulations! Your manuscript is now with our production department. 

Kind regards, 

on behalf of

Dr. Milkyas Endale 

Academic Editor

PLOS ONE